# Reliability of Extreme Wind Speeds Predicted by Extreme-Value Analysis

**Nicholas John Cook** 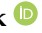

Independent Researcher, Highcliffe-on-Sea, Dorset BH23 5DH, UK; wind@njcook.uk

**Abstract:** The reliability of extreme wind speed predictions at large mean recurrence intervals (MRI) is assessed by bootstrapping samples from representative known distributions. The classical asymptotic generalized extreme value distribution (GEV) and the generalized Pareto (GPD) distribution are compared with a contemporary sub-asymptotic Gumbel distribution that accounts for incomplete convergence to the correct asymptote. The sub-asymptotic model is implemented through a modified Gringorten method for epoch maxima and through the XIMIS method for peak-over-threshold values. The mean bias error is shown to be minimal in all cases, so that the variability expressed by the standard error becomes the principal reliability metric. Peak-over-threshold (POT) methods are shown to always be more reliable than epoch methods due to the additional sub-epoch data. The generalized asymptotic methods are shown to always be less reliable than the sub-asymptotic methods by a factor that increases with MRI. This study reinforces the previously published theory-based arguments that GEV and GPD are unsuitable models for extreme wind speeds by showing that they also provide the least reliable predictions in practice. A new two-step Weibull-XIMIS hybrid method is shown to have superior reliability.

**Keywords:** generalized extreme value distribution; generalized Pareto distribution; Weibull distribution; Gringorten estimator; XIMIS; bootstrapping; epoch maxima; peak over threshold

---

## 1. Introduction

Comparing statistical parametric extreme-value (EV) models against observed field data is limited to showing how well an individual set of observations is estimated by the model's parameters in terms of goodness-of-fit. "Bootstrapping" [1], the sampling of many trials from a cumulative distribution function (CDF), allows models to be compared directly with the known source CDF and their statistical bias and variance to be determined. This is particularly effective for calibrating extrapolations, i.e., predictions for mean recurrence intervals (MRI) beyond the record length of the data. The choice of the source CDF depends on the scope of the calibration, e.g., an asymptotic model cannot be used to investigate asymptotic convergence as it is, by definition, fully converged. The datum epoch, $T$, for EV analysis of wind speeds, is conventionally taken as $T = 1$ year, i.e., annual maxima to enclose seasonal trends, but the available observational record may not provide sufficient maxima for analysis. This prompted the development of methods using sub-epoch maxima [2,3]. It is generally acknowledged that the annual rate of independent peak wind speeds from synoptic windstorms is typically $r \approx 150$ to 300 [2] and is lower for mesoscale events such as thunderstorm downbursts, so is never sufficient to achieve convergence. It follows that the CDFs of observed annual maximum wind speeds are sub-asymptotic, which has prompted considerable debate on the validity of the conventional asymptotic models and the development of penultimate models that capture the sub-asymptotic characteristics.

The debate has simmered through to the present, occasionally flaring into heated discussion and defense of published studies, as exemplified by the references and the Additional Bibliography provided in the Supplementary Material. The arguments are

based on EV theory and supported, or opposed, by appeals to observed or simulated wind data. They resolve into two opposing viewpoints:

1.  That the generalized extreme value distribution (GEV) for epoch maxima and its corollary for peak-over-threshold data, the generalized Pareto distribution (GPD), are suitable to represent extreme winds because they empirically fit the observed sub-asymptotic characteristics. The consequential prediction of a maximum upper limit to the wind speed is sometimes taken to be real (as incorporated into the Australian wind code) although no physical constraints exist to support the limit values.
2.  That GEV and GPD are unsuitable because:
    a.  GEV and GPD are asymptotic models, and *r* is too small [4] for convergence.
    b.  EV theory predicts that wind observations fall into the domain of attraction of the Gumbel distribution [5], which is unlimited in the upper tail.
    c.  A better GEV/GPD fit to observed or synthetic wind speeds is purely empirical and only valid within the fitted range. Extrapolations to MRI beyond the record length converge towards the wrong asymptote.

The sub-asymptotic XIMIS model [6] was developed to address these issues. It exploits the tail-equivalence of wind observations to the Weibull distribution by transforming the wind speed to obtain the fastest possible convergence to the Gumbel distribution [7].

While this author is convinced by the theoretical arguments of tail equivalence, domains of attraction, e.g., Theorem 3.15 in [5], and in the superiority of the sub-asymptotic XIMIS model over other methods, this present study sets all theoretical arguments to one side. Given that the principal purpose of EV models is to predict values with a given risk of exceedance, this study focuses solely on assessing the reliability of these predictions when values sampled from known sub-asymptotic source distributions are assessed by each method, i.e., by parametric bootstrapping [1].

## 2. Materials and Methods

### 2.1. Bootstrap Simulations

2.1.1. Source Distributions

The calibration study starts with the conventional presumption that the Weibull CDF:

$$P\{V\} = 1 - \exp\left(-\left(\frac{V}{C}\right)^w\right) \tag{1}$$

where $C$ is the scale and $w$ is the shape, is a suitable model to represent parent wind speeds [7]. Even if this were not the case, EV models should faithfully represent extremes sampled from this model. It follows that the sub-asymptotic Gumbel distribution (Type 1) model of [8] for tail-equivalent Weibull parents is the appropriate source for sampling values of epoch maxima from its CDF:

$$\Phi\{V\} = \exp(-\exp(-y_V)) \text{ and } y_V = (V^w - U^w)/D^w \tag{2}$$

where $V$ is the variate (wind speed), $U$ is the location (mode), $D$ is the scale (dispersion), $w$ is the Weibull shape parameter, and $y_V$ is the Gumbel [9] "reduced variate" modified to accommodate $w$. The transform of wind speed to $V^w$ gives tail-equivalence to the Exponential distribution so that extremes exhibit the fastest possible convergence to the Gumbel distribution [7]. Equivalence between the epoch and Weibull parameters is shown [8] to be:

$$U^w = C^w \ln(r) \text{ and } D^w = C^w \tag{3}$$

where $r$ is the rate of independent peaks per epoch.

### 2.1.2. Epoch Maxima

Simulated epoch maxima were rendered dimensionless by normalizing with the dispersion, rearranging (2) to:

$$V/D = (\Pi^w - \ln(-\ln(\Phi)))^{1/w} \tag{4}$$

and sampling for $\Phi$. Here, $\Pi = U/D$ is the dimensionless mode, the Gumbel [9] "characteristic product" and $\Pi^w = \ln(r)$, so increasing values of $\Pi$ and of $r$ both imply convergence towards the Gumbel asymptote.

### 2.1.3. POT Values

POT values were similarly made dimensionless on rearranging (1) to:

$$V/D = (-\ln(-\ln(P)))^{1/w} \tag{5}$$

sampling for $P$ to obtain a parent, then selecting the $N = rR$ highest values, where $R$ is the simulated record length in epochs. The POT threshold, therefore, corresponds to the $N$-th highest value. POT values are fundamentally different from epoch maxima, even when $r = 1$. Then the $R$ POT values are unevenly distributed among the $R$ epochs so that, on average 37% of the epochs will contain no value and the other 63% may contain more than one value, i.e., the 2nd or 3rd highest values in those epochs.

### 2.2. The Extreme-Value Models

### 2.2.1. Asymptotic Distributions

The cumulative distribution function (CDF), $\Phi\{V\}$, for the GEV is:

$$\Phi\{V\} = \begin{cases} \exp\left(-(1 + \xi y_G)^{-\frac{1}{\xi}}\right) & \text{if } \xi \neq 0 \\ \exp(-\exp(-y_G)) & \text{if } \xi = 0 \end{cases} \text{ and } y_G = (V - \mu)/\sigma \tag{6}$$

where $\mu$ is the location, $\sigma$ is the scale, $\xi$ is shape parameter, and $y_G$ is the dimensionless wind speed for the GEV. (The conventional symbol for GEV/GPD scale, $\sigma$, should not be confused with the standard deviation operator $\sigma(x)$ or $\sigma_x$.) The case where $\xi > 0$ is the Fréchet distribution (Type 2), unlimited in the upper tail, but limited to $\min(y_V) = -1/\xi$ in the lower tail. The case where $\xi < 0$ is the Reverse Weibull distribution (Type 3), unlimited in the lower tail but limited to $\max(y_V) = -1/\xi$ in the upper tail. The special case where $\xi = 0$ is the Type 1 or Gumbel distribution, which is unlimited in both tails.

The GPD is a corollary of the GEV for POT methods where the variate is the excess, $V - u$, over a threshold, $u$. Replacing $\mu$ with $u$ in (6) and keeping only the first-order term of the binomial expansion gives the usual GPD expression:

$$\Phi\{V - u\} = \begin{cases} 1 - (1 + \xi y_G)^{-\frac{1}{\xi}} \text{ if } \xi \neq 0 \\ 1 - \exp(-y_G) \text{ if } \xi = 0 \end{cases} \tag{7}$$

This assumes that all binomial terms after the first are negligible, requiring $y_G$ to be large, which makes the GPD an asymptotic approximation for the asymptotic GEV so, in a sense, it is doubly asymptotic.

### 2.2.2. Penultimate Distribution

The CDF of the general penultimate Type 1 distribution is given by (2), above. The distinction between epoch and POT data requires different mean plotting positions, $\overline{y}_V$, obtained from the order statistics of the samples.

For epoch maxima, estimators for $\overline{y}$ take the form:

$$\overline{y}_V = -\ln\left(-\ln\left(\frac{n - A}{R + 1 - A - B}\right)\right) \tag{8}$$

where *n* is the value rank from **smallest to largest**, [1..*R*]. The coefficient values $A = B = 0$ give the classic Weibull estimator which is biased for $\bar{y}_V$. The Gringorten estimator [10], $A = B = 0.44$ removes bias in $\bar{y}_V$ for very large samples. Cook and Harris [11] used bootstrapping to derive the sub-asymptotic coefficient values for finite *R* and, additionally, derived an expression for the statistical variance, $\sigma^2(y_V)$, to permit fitting by weighted least mean squares. This sub-asymptotic method is used here and referred to as "Gringorten-Cook-Harris", or "Gringorten" (GRG) for brevity.

For POT values, the estimators of $\bar{y}_V$ and $\sigma^2(y_V)$ for the XIMIS model [6] are:

$$\bar{y}_1 = \gamma + \ln(R) \text{ and } \bar{y}_{m+1} = \bar{y}_m - 1/m \tag{9}$$

$$\sigma^2_{y_1} = \pi^2/6 \text{ and } \sigma^2_{y_{m+1}} = \sigma^2_{y_m} - 1/m^2 \tag{10}$$

where *m* is the value rank in descending order, **largest to smallest**, $\gamma = 0.5772\ldots$ (Euler's constant) and *R* is the number of epochs in the sample. The values $\bar{y}_1$ and $\sigma^2_{y_1}$ are derived from asymptotic theory. Given that this study focuses on sub-asymptotic behavior, convergence towards these values was assessed by bootstrapping a range of *r* and *R*, each for $10^6$ trials. As expected, the values collapse against the population of extremes, *rR*, as shown in Figure 1, with the errors inversely proportional. The fitted equations can be used to correct the sub-asymptotic behavior but, as the errors are less than 1% for *rR* > 80, indicated by the dashed horizontal line, this is insignificant compared with the statistical variance of the variate.

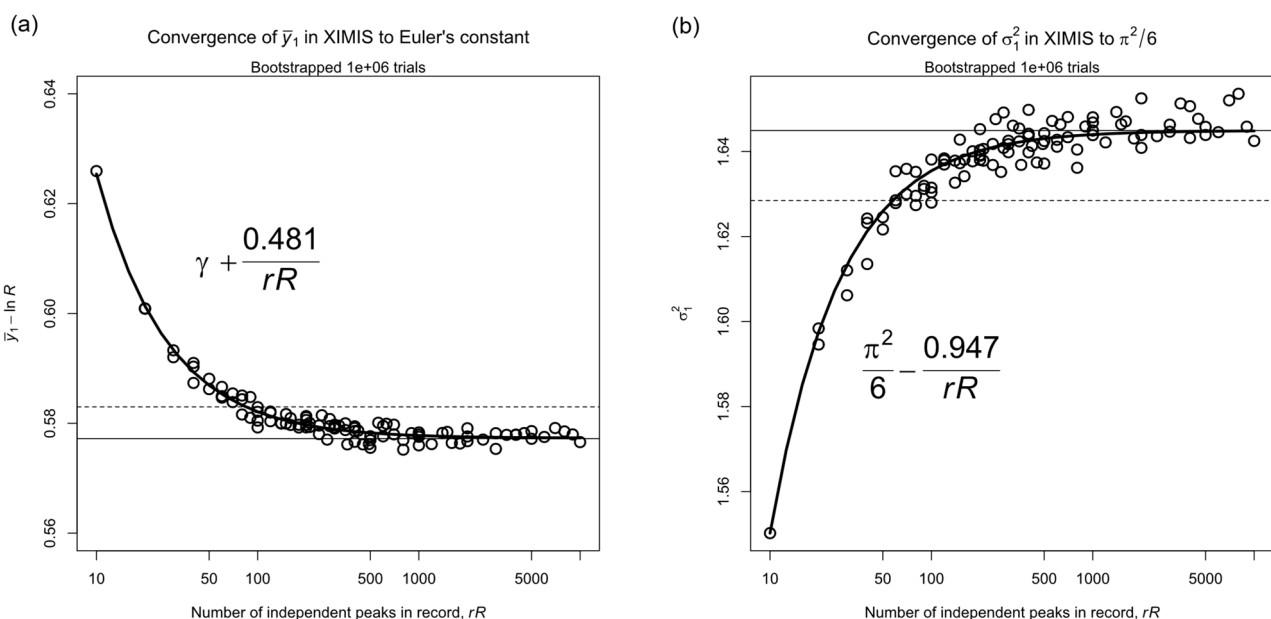

**Figure 1.** Asymptotic convergence of XIMIS reduced variate estimates for the highest value: (**a**) mean; (**b**) variance.

### 2.3. Some Example Model Fits

Some example fits of wind speed extremes sampled from typical parameter values are presented on the classical Gumbel axes in Figures 2 and 3, with the reduced variate as abscissa and the wind speed as ordinate. The Gumbel asymptote, $w = 1$, appears as a straight line on these axes, with intercept *U* and slope *D*. The penultimate and GEV/GPD models appear as a curve that is concave upwards when $w < 1$ or $\xi > 0$, and concave downwards when $w > 1$ or $\xi < 0$. For reasons of consistency of comparison and for stability when fitting a myriad of trials, weighted least-mean-squares (wLMS) as in [6] was used as the fitting method for all models, except for GPD which was fitted using probability-weighted moments.

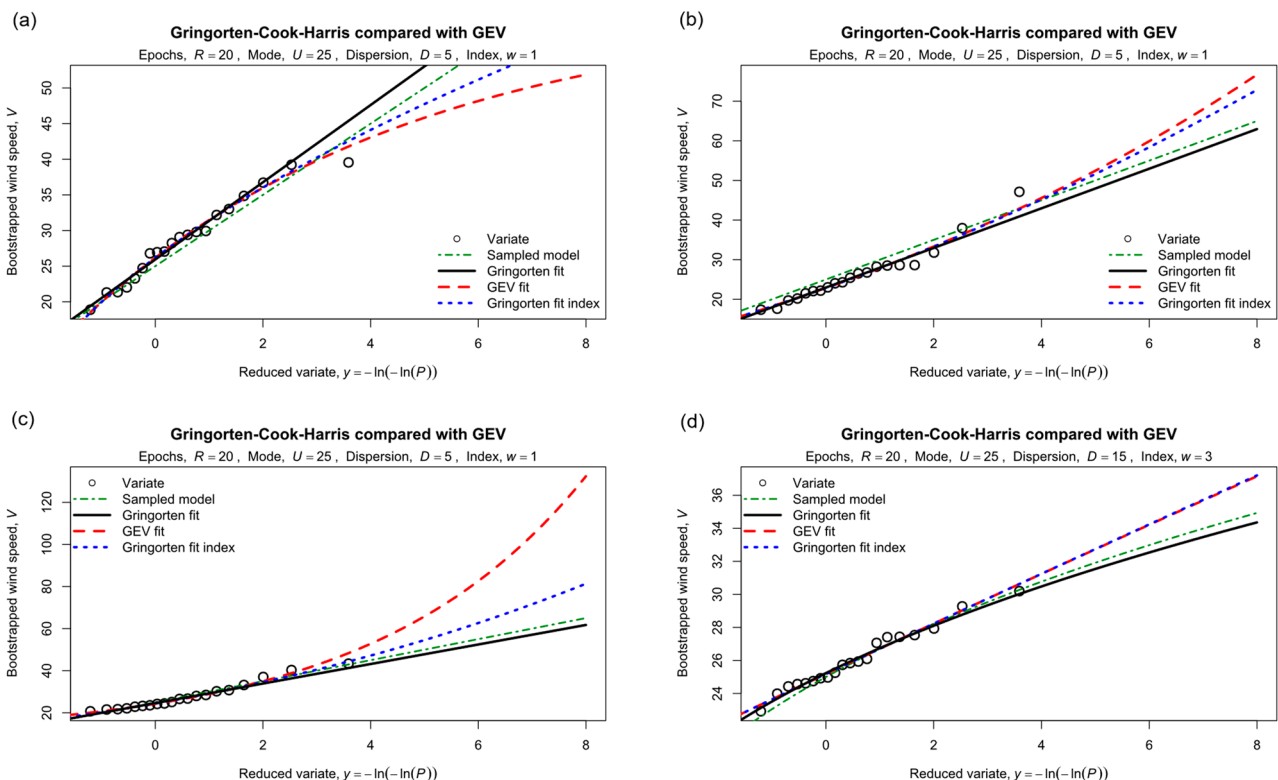

**Figure 2.** Four example trials fitted to epoch methods: "GEV fit"—fitting $U$, $D$ and $\xi$; "Gringorten fit"—with source shape, $w$, fitting 2 parameters, $U$ and $D$; and "Gringorten fit index"—fitting 3 parameters, $U$, $D$ and $w$.

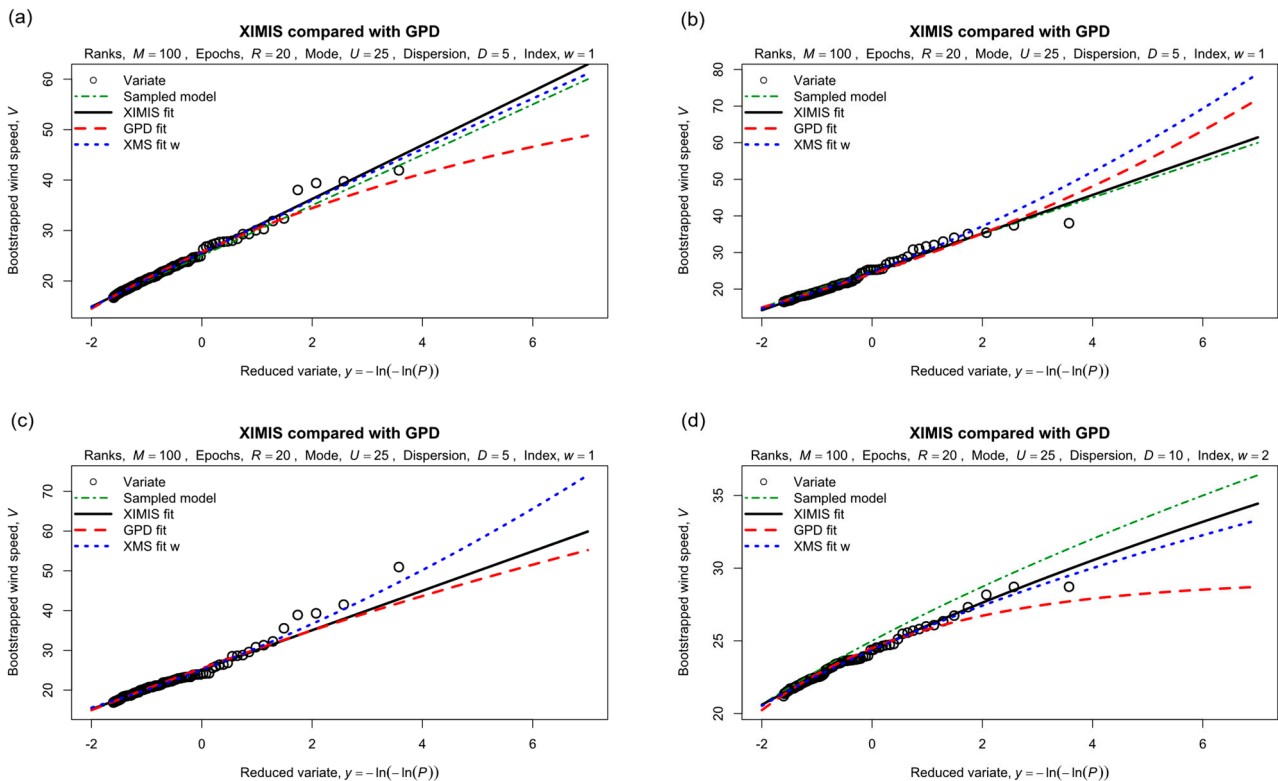

**Figure 3.** Four example fits to POT methods: "GPD fit"—fitting $D$ and $\xi$, with the threshold $u = V_M$, the lowest of the $M$ samples; "XIMIS fit"—with source shape, $w$, fitting $U$ and $D$; and "XMS fit $w$"—XIMIS fitting $U$, $D$ and $w$.

Figure 2a,b illustrates how the epoch methods in which the shape parameter is a free fit, together with location and scale, tend to follow any deviation of the samples away from the upper tail of the source. In the absence of tail deviations, Figure 2c illustrates the trend for the shape to follow curvature around the mode, in this case provoking a strong, spurious Fréchet response, $\xi > 0$, leading to unrealistically high predictions in the upper tail. Similarly, Figure 3a,b illustrates how the POT methods display the same trends, albeit less strongly, and with a different weighting between tail and body. Figure 3c shows that XIMIS tends to "chase the tail" while GPD follows the body curvature, for the same sample, when shape is a free fit, leading to shape estimates of an opposing trend.

The abovementioned examples are from asymptotic sources, $w = 1$, to demonstrate departures from the expected straight line, and in all these cases the Gringorten or XIMIS fits with the fixed source shape give close matches to the source distribution, illustrating the benefit of knowing the shape in advance. Figure 2d shows that the curvature of the source when $w = 3$, which is larger than any value observed in practice, is relatively small and that the variance of a small number of samples, here $R = 20$, will mask the curvature so that both methods underestimate the shape. On the other hand, Figure 3d shows that for $w = 2$, typical of temperate depressions [7], GPD can exaggerate the curvature, leading to an unrealistically low predicted value limit, whereas XIMIS remains consistent with the source.

## 3. Results

### 3.1. Bootstrap Trials: Phase 1

#### 3.1.1. Source Parameters

Bootstrapping was run for each combination of parameters of the sub-asymptotic source distribution (2). Three methods for epoch: GEV, Gringorten with source $w$ and Gringorten with free $w$; and three for POT: GPD, XIMIS with source $w$ and XIMIS with free $w$; were applied to each trial for direct comparison between methods. Common to all methods, the shapes and characteristic products trialed ranged in increments from: $w = 0.75 \rightarrow 4$, and $\Pi = 1.5 \rightarrow 15$. For the epoch methods only, the epochs trialled were: $R = 10 \rightarrow 100$ in increments of 10, For the POT methods only, the number of sample values above the threshold were: $M = 50, 75, 100, 125, 150$ and $200$, each for epochs $R = 20$ and $50$. These ranges span those typically found in hourly wind speed observations.

For each combination of source parameters $10^4$ bootstrap trials were run, the parameters fitted for each model, and predictions for MRI = 50, 100, 1000, and 10,000 were compiled. The ensemble means and variances of the model parameters and predictions are supplied as Rdata files in the Supplementary Material.

#### 3.1.2. Expectation of Results

As the bootstrap was rendered dimensionless by the dispersion, $D$, of the source distribution, the expectation for the ensemble means, $\langle \cdot \rangle$, of a perfect model is that the mode $\langle U \rangle \rightarrow \Pi$, the dispersion $\langle D \rangle \rightarrow 1$, the shape $\langle w \rangle \rightarrow w$ and the sample prediction of wind speed for a given MRI to $\langle V/D \rangle = (\Pi^w - \ln(-\ln(1 - 1/\mathrm{MRI})))^{1/w}$. As the mode is not a bootstrap parameter of the GPD, the corresponding values are evaluated at $y = 0$.

#### 3.1.3. Reliability of Predictions

This study focuses on the reliability of predicted wind speed at MRI, which is relevant for the design of structures. Most codes of practice for buildings base their characteristic, or "basic" [12], design values on 50 years. For bridges and other vital infrastructure, the value is around 100 years. However, the application of safety factors raises the notional MRI for significant structural damage to around 1000 years and for collapse, serious injury, or death to around 10,000 years. MRI is often equated to the "design life" of a structure, whereas, in practice, it is the risk of exceedance in each year a structure is exposed to the wind that is relevant, however long its design life might be. Hence, the MRI values adopted in this

study: 50, 100, 1000, and 10,000 years correspond to an annual risk of exceedance of 0.02, 0.01, 0.001, and 0.0001, respectively.

The multiple parameters of this study make reporting all combinations impracticable. The most critical parameters are $w$ and $\Pi$ for the source distribution which together dictate the degree of convergence of the models towards the asymptote and their curvature when plotted on Gumbel axes. The predictions are presented as contour plots on the $w$–$\Pi$ plane for: MRI = 50 and 10,000 with $R = 50$ for epoch methods; and $M = 100$ with $R = 20$ for POT methods. Reliability is expressed in terms of the bias error, the predictable error component, and the standard error, the random error component.

Bias Errors

The mean bias error for a given MRI is defined as:

$$\bar{\varepsilon}_{\text{MRI}} = (\langle V_{\text{MRI}} \rangle - V_{\text{MRI}}) / V_{\text{MRI}} \tag{11}$$

where $\langle V_{\text{MRI}} \rangle$ is the ensemble mean of the predictions from all trials and $V_{\text{MRI}}$ is the exact value from the source distribution, i.e., the error expressed as a fraction of the true value from the source distribution.

Figures 4 and 5 present the mean bias error for MRI = 50 and 10,000, respectively. All methods give very small bias errors at MRI = 50, exceeding 1% only where $w < 1$ and $\Pi$ is small. However, at MRI = 10,000, the GEV and GPD bias errors exceed +10% for $w < 1$ due to the potentially high values from spurious Fréchet responses, as in Figure 2c. A pattern is evident in the small variations across the $w$–$\Pi$ plane for Gringorten and XIMIS fitting $w$ (centre column) and for GEV and GPD (right-hand column) which is consistent in each case.

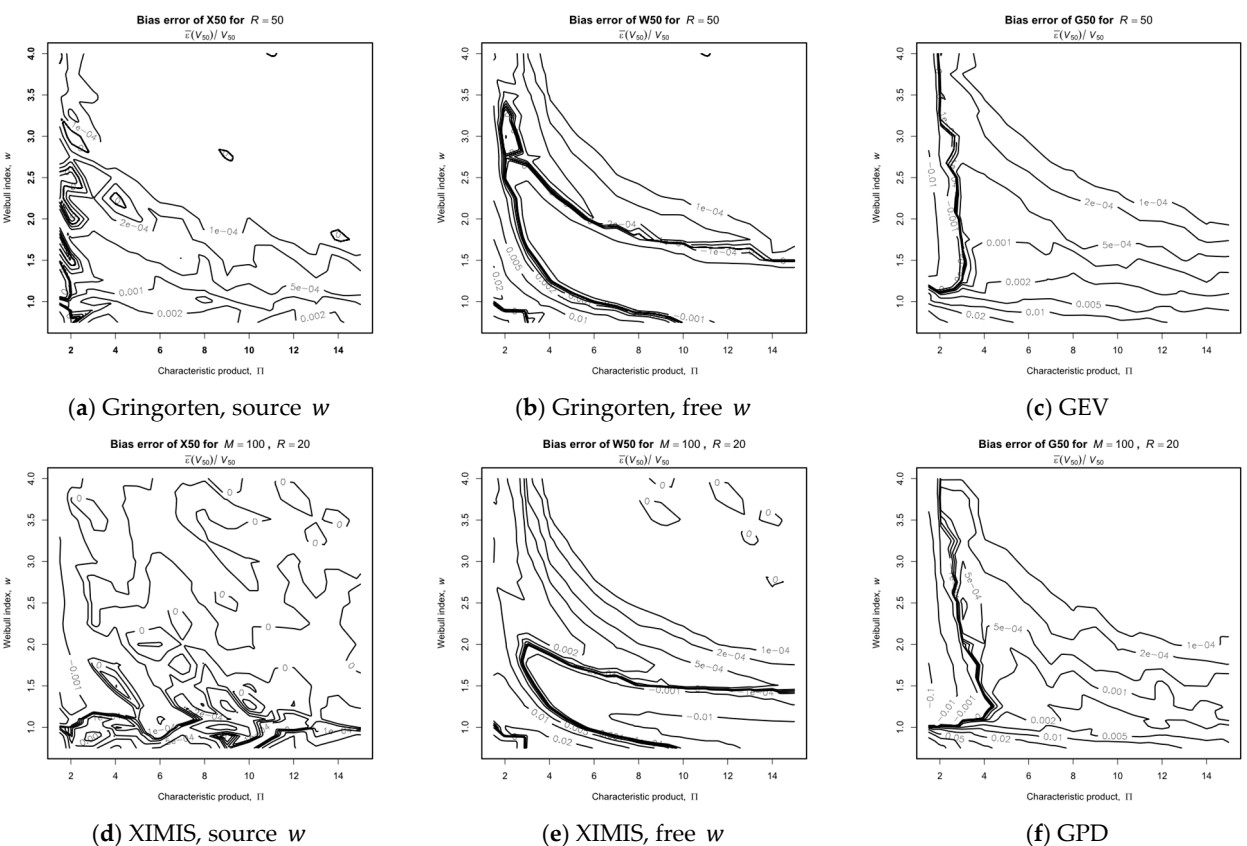

**Figure 4.** Mean bias error for MRI = 50: (**a**–**c**) Epoch methods for $R = 50$; (**d**–**f**) POT methods for $R = 20$ and $M = 100$.

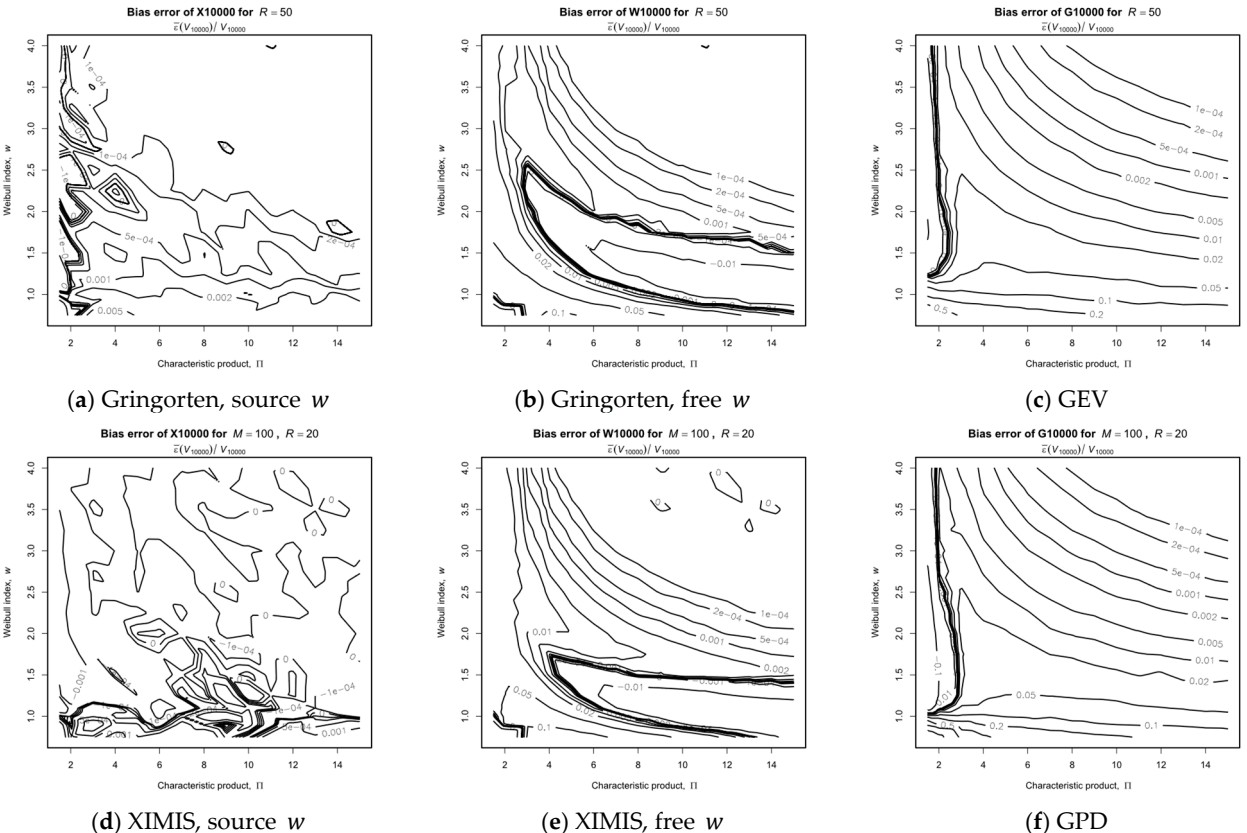

**Figure 5.** Mean bias error for MRI = 10,000: (**a**–**c**) Epoch methods for $R = 50$; (**d**–**f**) POT methods for $R = 20$ and $M = 100$.

Standard Errors

The standard error for a given MRI is defined as:

$$\sigma_{\mathrm{MRI}} = \sqrt{\left\langle \sigma^2_{V_{\mathrm{MRI}}} \right\rangle} / V_{\mathrm{MRI}} \tag{12}$$

where $\left\langle \sigma^2_{V_{\mathrm{MRI}}} \right\rangle$ is the ensemble variance of the predictions from all trials and $V_{\mathrm{MRI}}$ is the exact value from the source distribution, i.e., the standard deviation expressed as a fraction of the true value from the source distribution.

Figures 6 and 7 present the standard errors for MRI = 50 and 10,000, respectively. In all cases, the standard errors are an order of magnitude greater than their respective bias errors, indicating that variability of the predicted values is the primary concern. The contour pattern is consistent across all methods, with the variability decreasing with increasing $w$ and $\Pi$. For the penultimate Gringorten and XIMIS models, there is an advantage to knowing the shape, $w$, and so fitting only for $U$ and $D$ (left-hand column), as the standard error doubles when the shape is also fitted (center column). At the design MRI = 50, Figure 6, the GEV/GPD standard error (right-hand column) remains similar to the Gringorten and XIMIS free fit to $w$. However, at MRI = 10,000 for collapse, serious injury or death, Figure 7, the GEV/GPD standard error is much greater, reaching 50% for $w = 1$ at all $\Pi$ and exceeding 200% for $w = 0.75$ and $\Pi = 2$ (unlikely to be encountered in practice).

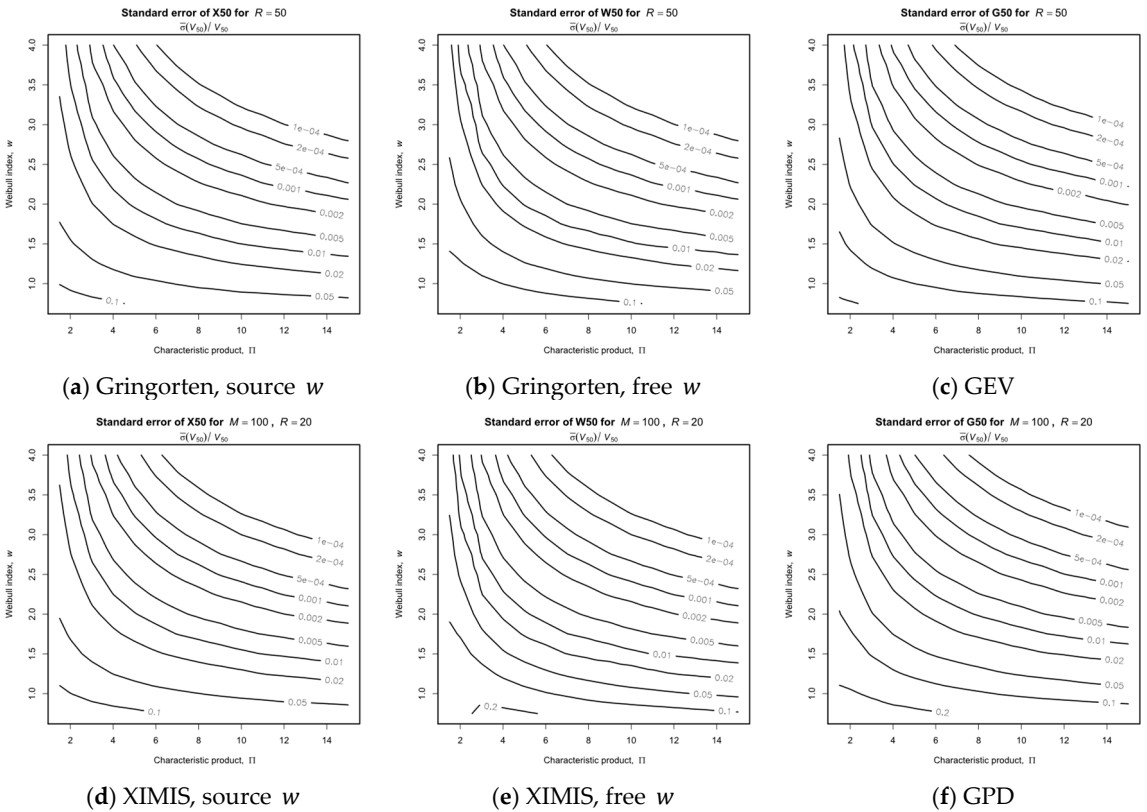

**Figure 6.** Standard error for MRI = 50: (**a**–**c**) Epoch methods for $R = 50$; (**d**–**f**) POT methods for $R = 20$ and $M = 100$.

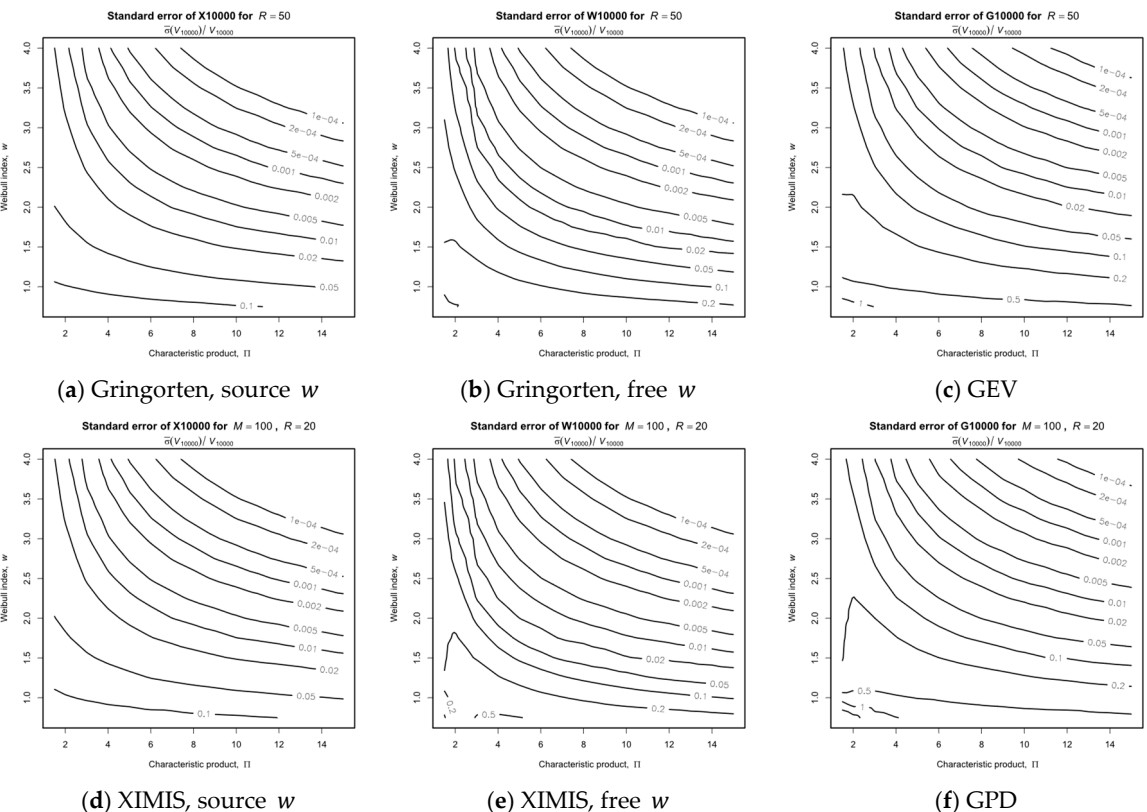

**Figure 7.** Standard error for MRI = 10,000: (**a**–**c**) Epoch methods for $R = 50$; (**d**–**f**) POT methods for $R = 20$ and $M = 100$.

### 3.1.4. Performance Overview

The examples in Figures 4–7 demonstrate the performance of the methods for typical parameter combinations. They show the significant advantage of knowing the shape in advance and fitting only for location, *U*, and scale, *D*. They also demonstrate that the variability of the predictions is an order of magnitude greater than the mean bias, so that standard error is the critical metric for judging overall performance.

The quantile–quantile (QQ) plot provides a simple way to compare the reliability of methods on a single chart. Figure 8 presents the QQ plots for *R* = 20 epochs, comparing the standard error from the three-parameter fits at mean recurrence intervals MRI = 50 and 10,000 against the corresponding datum Gringorten or XIMIS two-parameter fit with source *w*. For clarity, the datum is shown by the straight line of slope 1 through the origin as all two-parameter fit values collapse onto this line. The standard errors for the generalized methods, GEV and GPD, are consistently worse than for the penultimate Gringorten and XIMIS methods, and both are worse than the datum.

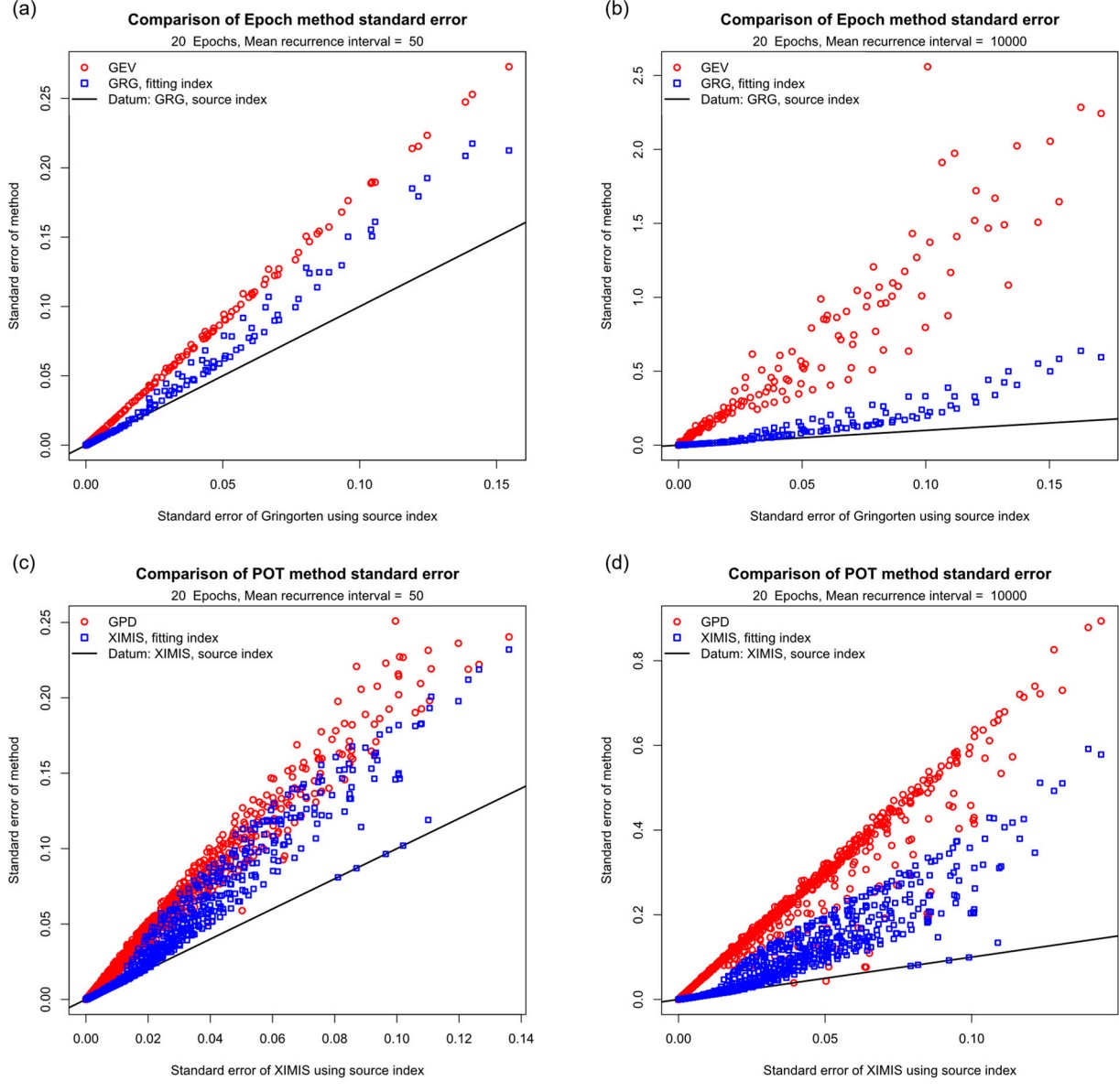

**Figure 8.** Quantile-quantile plots of standard errors for epoch and POT methods, against the known shape datum, *R* = 20: (**a**) Epoch, MRI = 50; (**b**) Epoch, MRI = 10,000; (**c**) POT, MRI = 50; (**d**) POT, MRI = 10,000.

Finally, the overall performance of all the models is compared in Figure 9 by the prediction standard error for $R = 20$ epochs, ensemble averaged for all $w$ and $\Pi$. The $\sigma_{V/D}$ values for each MRI are shown sorted by increasing error for each of the datum recurrence intervals. For clarity, the ordinate scale is expanded and clipped at $\sigma_{V/D} = 3$, the GEV value of $\sigma_{V/D} = 9.08$ at MRI = 10,000 exceeding this by a factor of three. The penultimate methods where $w$ is known, XIMIS followed by Gringorten, are always best and the generalized methods, GPD and GEV, are always worst; with the disparity increasing with higher MRI. The performance of the penultimate methods where $w$ is fitted always lies in between.

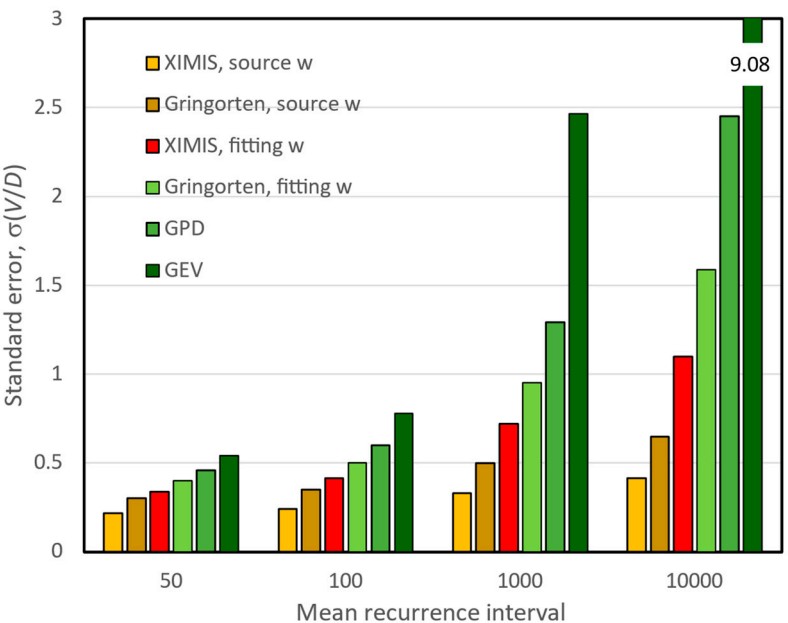

**Figure 9.** Prediction standard errors for each method at datum mean recurrence intervals, for $R = 20$ years ($M = 100$ for POT methods).

### 3.1.5. Shape Parameter

Having demonstrated the superior reliability of the penultimate methods over the generalized asymptotic models, the behavior of the fitted shape parameter, $w$ or $\xi$, is examined in more detail. In the penultimate models, the Weibull index, $w$, preconditioned the transformed wind speed, $V^w$, for fastest convergence. In the generalized asymptotic models, $\xi$ mimicked the sub-asymptotic curvature in the body of the distribution but was seen to become progressively unreliable into the upper tail.

When either shape parameter is estimated along with the location and scale by a three-parameter fit its value is determined from the curvature of the plots, as in Figure 2. The curvature decreases towards zero with increasing $\Pi$ as the source distribution converges towards the linear Gumbel asymptote, so that a fit for $w$ or $\xi$ becomes increasingly ill-conditioned [8]. The corollary of this is that predicted values become less sensitive to the shape as $\Pi$ increases.

The fitted shape parameters corresponding to Figures 4–7, are presented in Figure 10. The contours of fitted $w$ for the penultimate models, (a) epoch and (d) POT, should present as horizontal lines but instead form a characteristic pattern that echoes the faint pattern in the corresponding mean bias errors in Figures 4 and 5. The contours of fitted $\xi$ for the generalized models, (b) GEV and (e) GPD match the pattern theorized in [8], and the lack of any concomitance between $w$ and $\xi$ is evidenced by the failure of the quantile-quantile (QQ) plots (c) and (f) to collapse.

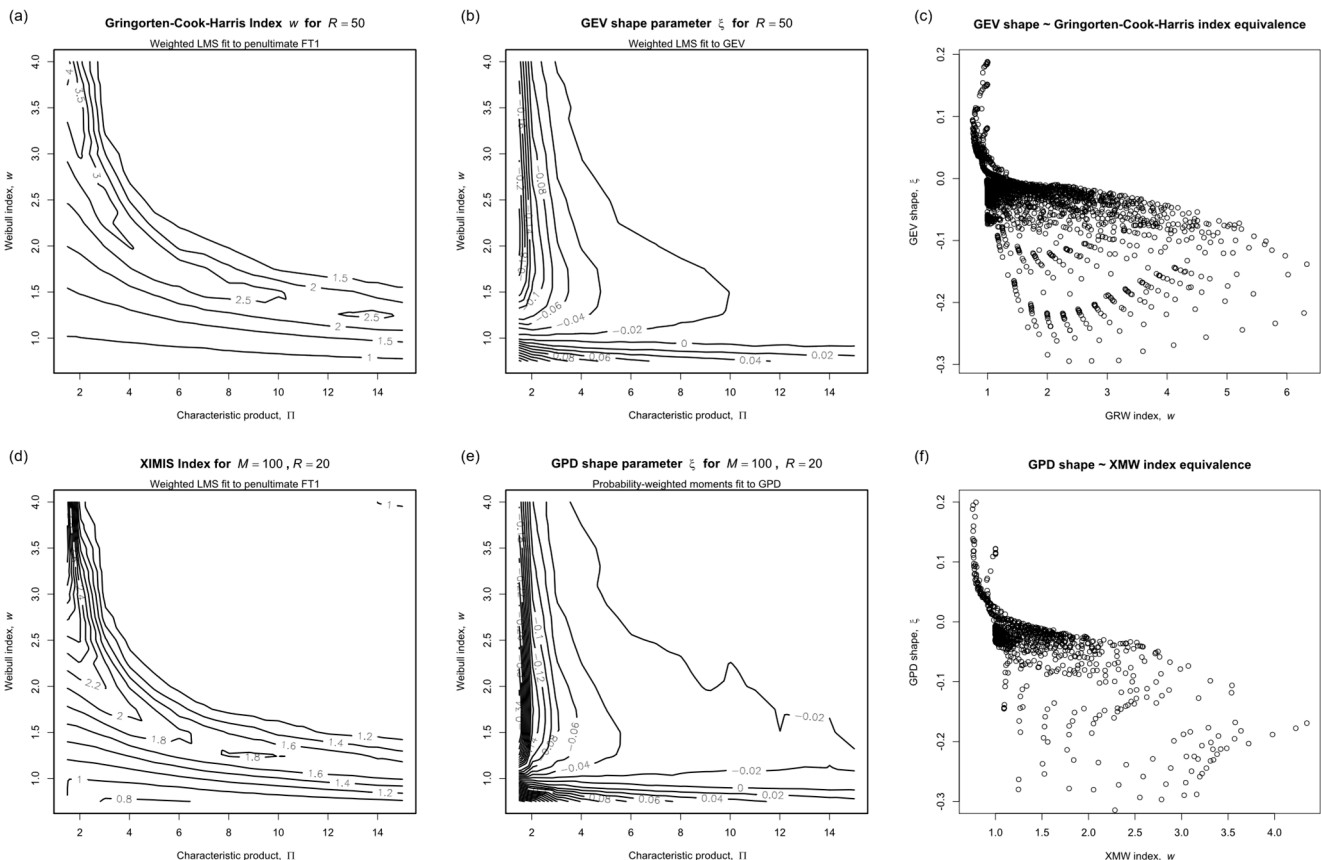

**Figure 10.** Fitted shape parameter: (**a**–**c**) Epoch methods for $R = 50$; (**d**–**f**) POT methods for $R = 20$ and $M = 100$.

The source value of $w$ is known in this study only because the bootstrap samples were extracted from a known distribution. In the practical case where wind speed observations are the source, $w$ is not known directly and must be determined from prior knowledge or by a suitable analysis of the data. The next phase of this study explores the issues involved in obtaining a suitable value from parent observations.

### 3.2. Bootstrap Trials: Phase 2

#### 3.2.1. Preamble

Wind speeds are generated by a variety of different causal mechanisms acting exclusively (disjoint) under changing synoptic conditions at different latitudes, producing parent distributions of all wind speeds that are disjoint mixtures of individual distributions of different scales. The recently developed Offset Elliptical Normal Mixture (OENM) model [13] predicts that most wind climates, worldwide, resolve as mixtures of bivariate Normal components when expressed as orthogonal vectors instead of speed and direction. This leads to the expectation that $w \approx 2$ for each component, typically in the range $w = 1.8 \rightarrow 2.2$, so that dynamic pressure $\propto V^2$ converges faster to the Gumbel asymptote than speed, $V$, [2].

Of the most relevance in EV analysis is the component that dominates the upper tail, e.g., Zhang et al. [14] report that thunderstorm downbursts are the dominant component in the mid-latitude mixed climate of Italy. However, determining the characteristics of this relevant component is not straightforward due to the influence of the other components. The parent distribution of a disjoint mixture, $P$, is given by the sum of the individual component distributions, $P_i$, weighted by their relative frequencies, $f_i$:

$$P = \sum f_i P_i \tag{13}$$

where $0 \leq f_i \leq 1$ and $\sum f_i = 1$. As the distribution, $P_1$, of the strongest component becomes dominant in the upper tail:

$$P \rightarrow f_1 P_1 + (1 - f_1) \quad \text{as } V \rightarrow \infty \tag{14}$$

Consider the case of a mixture of two Weibull distributions with the same shape, $w = 2$, but with different scales, $C$. The Weibull plot for $C_1 = 25$, $C_2 = 10$, $f_1 = 0.4$ and $f_2 = 0.6$ is shown in Figure 11a. The circles representing $P$, evaluated at integer values of $V$, are transitional between the two component distributions, $P_1$ and $P_2$, and show a good linear fit to $w = 1.3$. This is very much lower than the $w = 2$ of the individual components. The $w$ values obtained from a mixed parent will always be too low and so be inconsistent with those derived from processing extremes, as demonstrated by Torrielli et al. [15] who quote values in the range $w = 1.4 \rightarrow 1.7$ in Liguria, and as low as $w = 1.16$ elsewhere Italy. Figure 11b demonstrates how the fitted value varies with relative frequency, $f_1$, and the disparity in $C$, indicating that the greatest dilution occurs when the dominant component is strong and rare. Almost any departure from the independent and identically distributed (iid) random properties, including seasonal variation of a single component, will result in similar dilution of the fitted value of $w$.

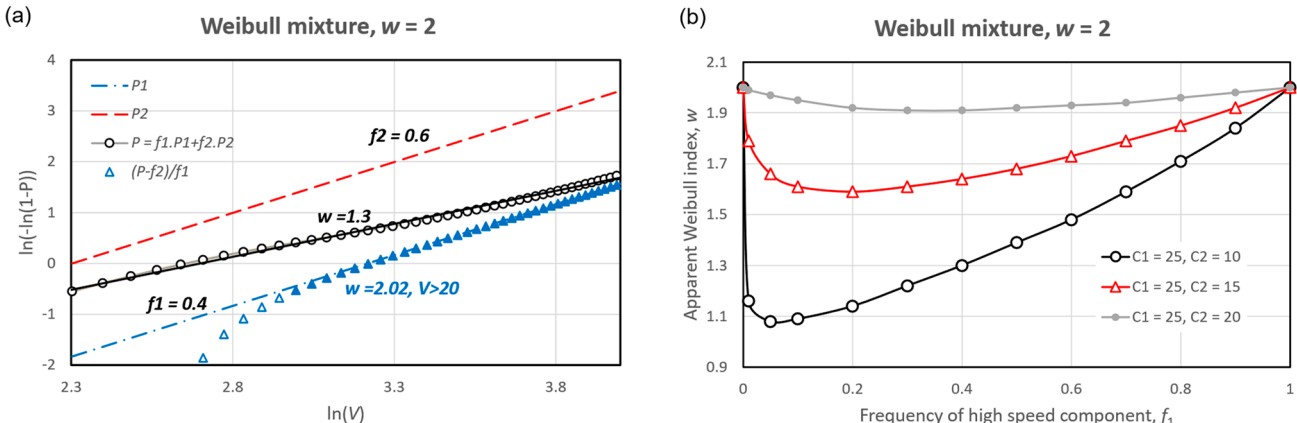

**Figure 11.** Weibull index underestimation of mixed distributions: (**a**) Typical fit to two-component mixture; (**b**) Variation of apparent index, $w$, with relative frequency, $f$.

If the relative frequency, $f_1$, can be reliably estimated, e.g., by counting thunderstorms, $w$ can be estimated from the convergence to (14) in the upper tail, as shown by the triangle symbols in Figure 11a. A linear fit for $V > 20$, indicated by the filled triangles, estimates $w$ to 1% accuracy.

Gomes and Vickery [16] revealed the need to extract the individual components of a mixture for separate analysis of extremes, and this separation also greatly simplifies the analysis of their relevant parents. However, separation usually includes a selection process, such as:

1. Left censoring a mixture at a threshold high enough to exclude all but the dominant component.
2. Separation of peak values by a minimum time increment [3].
3. Selecting peak values from independent synoptic events [2].

Selection of peak values moves the POT distribution slightly away from the original parent and towards the sub-asymptotic extreme distribution but preserves Weibull equivalence in the upper tail. The aim, therefore, in analyses of observations is to estimate the value of $w$ which provides the preconditioning transformation $V^w$ that best represents the sub-asymptotic Type 1 model.

### 3.2.2. Peak over Threshold Observations

The abstraction of POT data usually implies access to the full parent observations allowing $w$ to be estimated directly from the Weibull model (1), e.g., as the slope when plotted on Weibull axes: $\ln(-\ln(1 - P))$ as ordinate against $\ln(V)$ as abscissa. The Gringorten [10] estimator for $P$ used in (8) also removes bias from $\ln(1 - P)$ so that:

$$P = 1 - (m - 0.44)/(N + 0.12) \quad \text{where } N = rR \tag{15}$$

Equation (15) requires the parent population, $N$, which is unknown for left-censored POT observations unless it can be independently estimated. This is a common dilemma in lifetime analysis studies when failures occur at some time before the first inspection. The usual solution is to find the value of $N$ that gives the best linear fit, by minimizing the residual standard error or by maximizing likelihood [17].

The smallest possible value of $N$ is the population of left-censored POT data while the largest possible value is half the population of the parent (because, to qualify as a peak, there must be at least one lower value before the next peak). This defines the bounds for optimizing $N$ by Brent's method. Figure 12 shows examples of optimizing $N$ for the minimum standard error of a linear fit using the top $M = 100$ ranks for $w = 1, 2$ and 3. The solid line represents the source Weibull distribution. The (black) circles show the left-censored sample for the initial value $N = rR = 2000$, typical of a 20-year record of synoptic data. A linear fit to these would seem reasonable but it overestimates $w$. The (red) triangles show the optimized distribution and the optimal $w$ value is given in the chart heading, together with its percentage error.

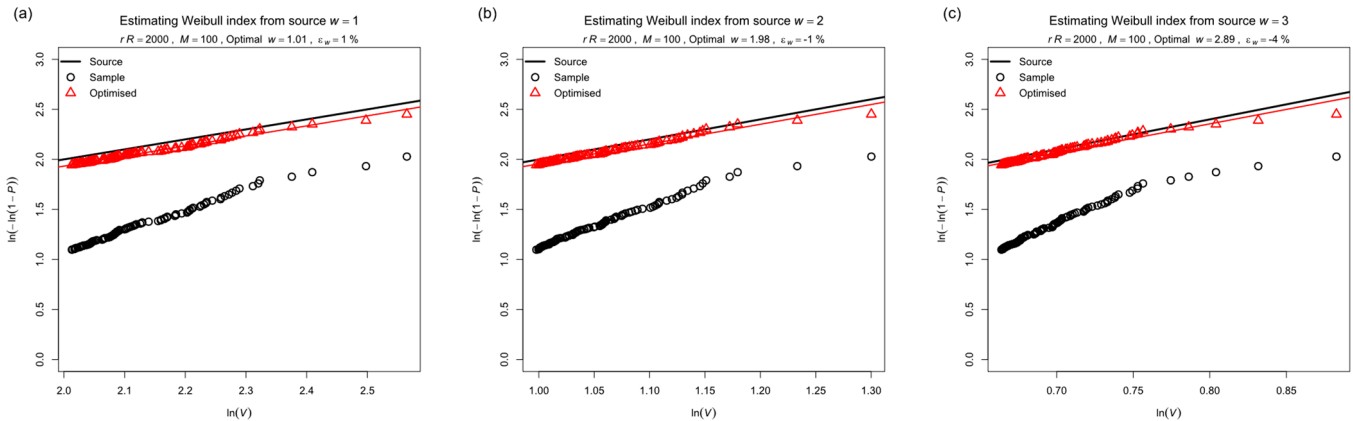

**Figure 12.** Examples of fitting a Weibull distribution to the upper tail by optimizing $N$ for minimum residual standard error: (**a**) Source $w = 1$; (**b**) Source $w = 2$; (**c**) Source $w = 3$.

### 3.2.3. Reliability of Shape Parameter Estimates

The results of running $10^4$ bootstrap trials of Weibull distributions for each source $w$, as before, for $r = 5 \rightarrow 500$ in increments with $R = 20$ and 50, then fitting the top $M = 100$ ranks are shown in Figure 13. $M = 100$ corresponds to 5 observations per year over the 20-year record, so is typical of the minimum POT population expected from observed records and therefore represents a lower bound of reliability. The ensemble mean, $\langle w \rangle$, and standard error, $\langle \sigma_w \rangle$, of all trials were compiled for each combination of $w$, $r$ and $R$, and are provided in the Supplementary Information files.

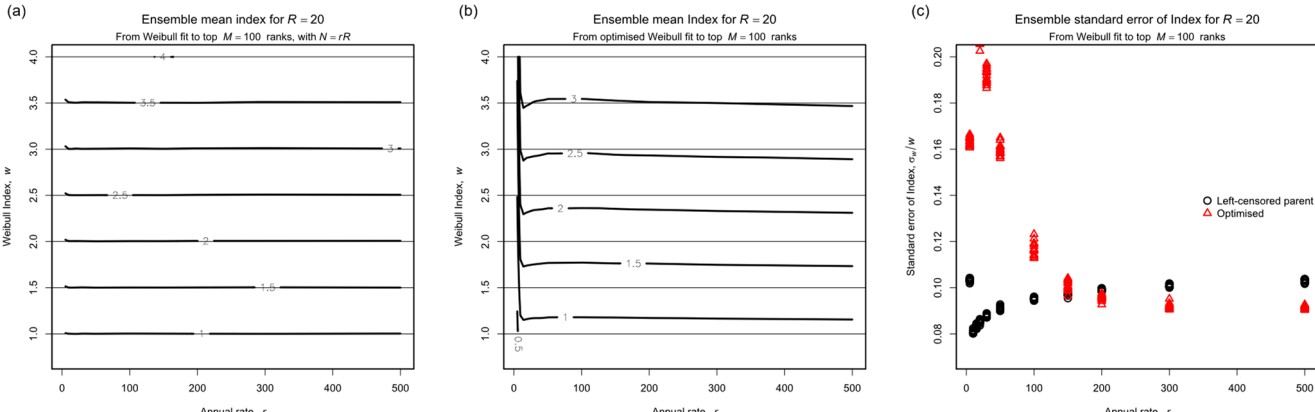

**Figure 13.** Weibull distribution index: (**a**) ensemble mean $w$ for left-censored parent; (**b**) ensemble mean $w$ optimised from top 100 ranks; (**c**) corresponding ensemble standard errors.

Figure 13a presents $\langle w \rangle$ compiled using $N = rR$, i.e., the POT population. This assumes that the POT threshold has been optimized to maximize the population of independent values when many may be excluded, in practice, so that this $N$ is an underestimate. Nevertheless, the contours in (a) present as horizontal lines indistinguishable from the source $w$ ordinate, so there is no discernible bias in $\langle w \rangle$. As earlier, the principal concern is the variability in terms of the standard error, which collapses in (c) with $\sigma_w / w$ as ordinate for all values of $w$ and remains <10%.

Figure 13b presents $\langle w \rangle$ as optimized by the process shown in Figure 12 when $N$ is unknown. The contours again present as near horizontal lines but indicate a consistent underestimate of ~15%. The corresponding standard error in (c) falls quickly from ~20% to below 10% for $r > 150$. Contrast Figure 13a,b with $\langle w \rangle$ from the three-parameter EV fit in Figure 10b. Also note in (b) the anomalous behavior of $w$ optimized at $r = 5$, corresponding to the special case of $rR = M$.

### 3.2.4. Standard Errors for XIMIS Preconditioning Options

Preconditioning XIMIS is a two-stage process whereby the value of the shape, $w$, is predetermined so that XIMIS reverts to the two-parameter fit of location, $U$, and scale, $D$. As this study eschews theory to focus on the practical analysis of the POT data, the two available options are those illustrated in Figure 13a,b. Namely:

1. Fitting the top $M$ values to the Weibull distribution where $N$ in (15) is the POT population, $N = rR$, or has been directly counted by identifying independent events.
2. Optimizing for the value of $w$ that gives the best fit to the Weibull distribution.

XIMIS preconditioned in this manner is denoted by Weib-XIMIS.

Bootstrap trials were run for both options for $M = 100$ and excluding the anomalous cases when $r = 5$. Standard errors grow rapidly with decreasing values of $M < 100$, while values of $M > 100$ show little improvement, indicating close convergence to the sample variance. The resulting QQ plots are presented in Figure 14 for MRI = 50 and 10,000. For comparison with Figure 8, the corresponding values from Figure 8c,d, where $w$ was fitted by XIMIS with $U$ and $D$, are shown by the small grey circles in the background.

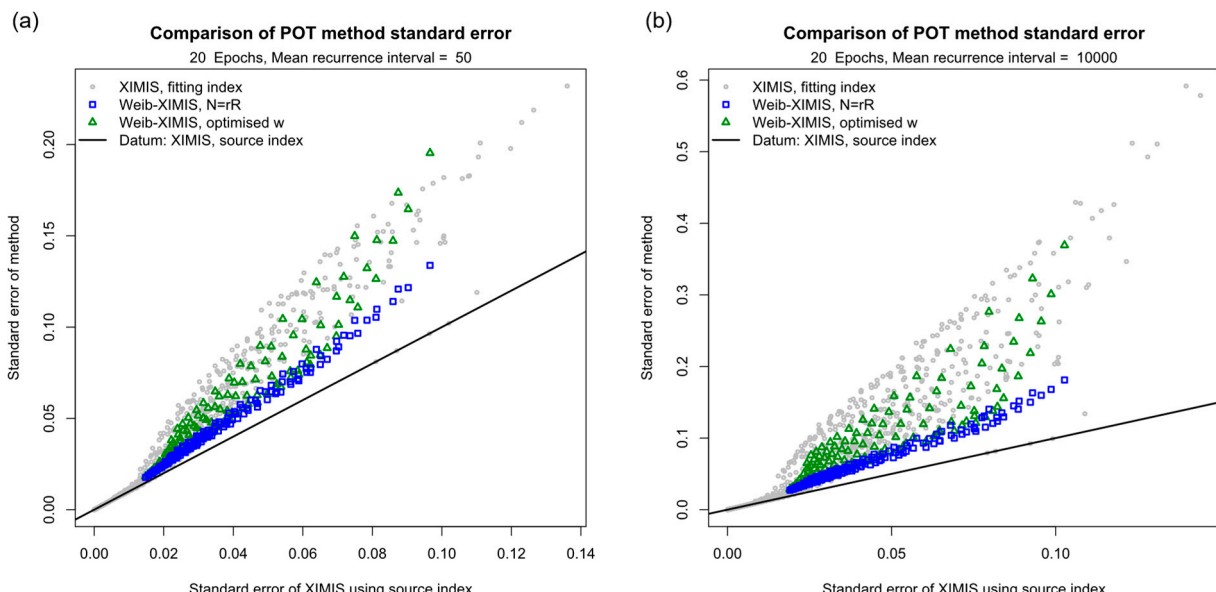

**Figure 14.** Quantile-quantile plots of standard errors for the XIMIS preconditioning options, against the source shape datum, $R = 20$: (**a**) MRI = 50; (**b**) MRI = 10,000.

The standard errors for the Weib-XIMIS predictions obtained by optimizing $w$ are shown by the triangle symbols. By limiting the fit to only the top $M = 100$ values, the sparser points reveal a grid pattern where values from right to left represent increasing rate, $r$, and values from top to bottom represent increasing $w$ for the source distribution. The standard errors occupy the same range as for the XIMIS three-parameter fit, which should not be a surprise since both operate on second derivatives: XIMIS by evaluating the curvature on the Gumbel plot and Weib-XIMIS by eliminating curvature on the Weibull plot.

The standard errors for the Weib-XIMIS predictions obtained with $N = rR$ are shown by the square symbols. These resolve as a narrow linear band under the bottom edge of the scatter for the other two XIMIS options, confirming this to be the most reliable method. The remaining difference between this and the datum represents the cost of needing to estimate the unknown shape parameter, $w$.

### 3.2.5. Characteristic Product

As the Phase 2 trials required the rate, $r$, in place of the characteristic product, $\Pi$, for generation of the Weibull parent source, the opportunity arises to confirm that the relationship $\Pi^w = \ln(r)$ expected from theory survives the application of XIMIS. Figure 15 shows how $\Pi^w$ evaluated from the XIMIS estimates for $U$, $D$ and $w$, collapses for all source $w$ when plotted against $\ln(r)$. Where $w$ in XIMIS is the source value, denoted by the circles, the collapse is indistinguishable from the expectation. Where $w$ is predetermined from the POT population, $N$, the collapse is good but $\Pi^w$ is marginally (3%) overestimated. Where $w$ is optimized from the top $M = 100$ ranks the collapse is not complete and $\Pi^w$ is overestimated by 15%.

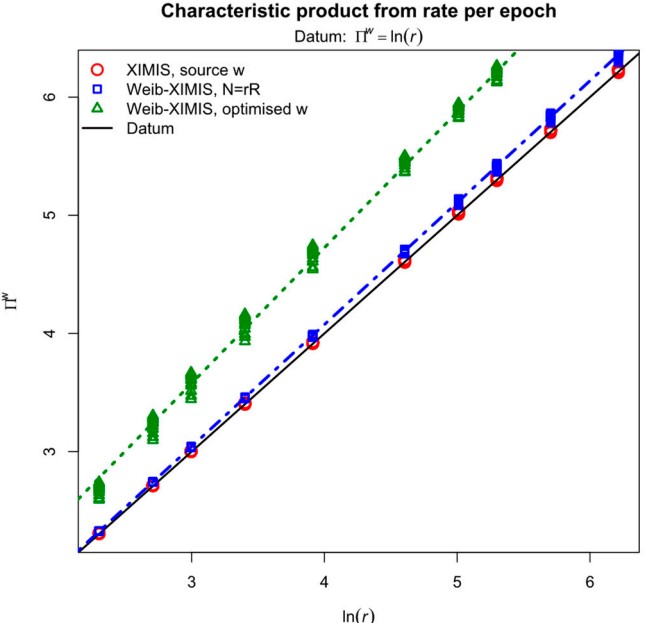

**Figure 15.** Relationship between the characteristic product from XIMIS and the source rate per epoch.

3.2.6. Sensitivity of XIMIS Predictions to the Shape Parameter, $w$

Finally in this phase, the reducing sensitivity of the predictions to the shape parameter with increasing $\Pi$, noted earlier, was investigated by bootstrapping Weib-XIMIS for $M = 100$ and applying various percentage errors to the value of $w$ used in the fit. Figure 16 shows the resulting percentage errors in the predictions for three key parameter combinations. In all three cases the prediction errors reduce with increasing $\Pi$, as expected, and are of the opposite sign to the error in $w$, i.e., the predictions are underestimated if $w$ is too large and overestimated if too small. For $w = 2$, the value typical of temperate depressions, the prediction errors in (b) for MRI = 10,000 are not much larger than in (a) for MRI = 50 and decrease by a factor of 100 over the range of $\Pi$. For $w = 1$, which has not been observed so represents a lower limit, the prediction errors decrease by the smaller factor of 10. For larger values of $w$ the prediction errors decrease by ever larger factors: effectively by $10^{-w}$.

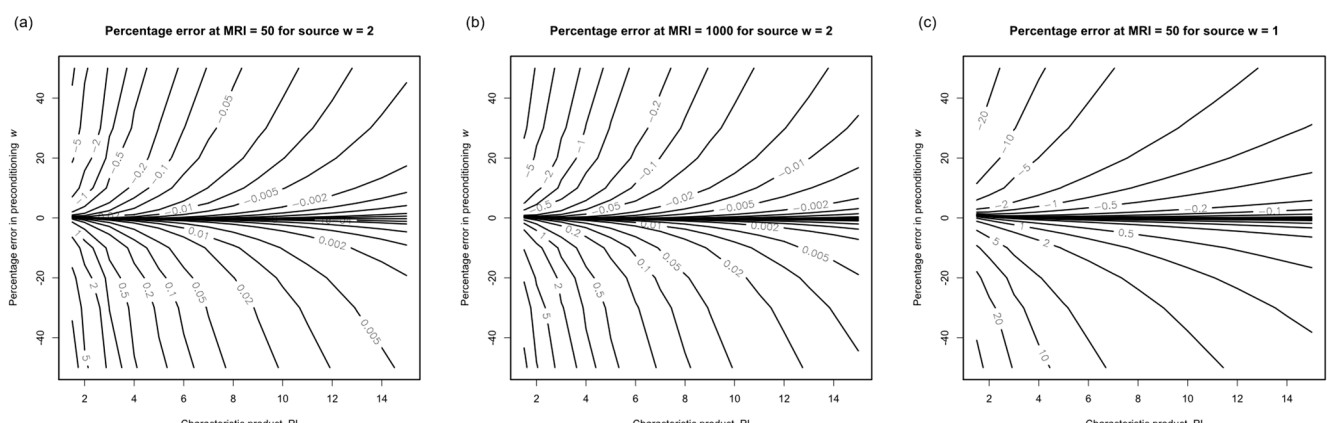

**Figure 16.** Sensitivity of Weib-XIMIS predictions to error in the preconditioning shape, $w$: (**a**) $w = 2$, MRI = 50; (**b**) $w = 2$, MRI = 10,000; (**c**) $w = 1$, MRI = 50.

*3.3. Bootstrap Trials: Phase 3*

3.3.1. Preamble

The Phase 1 and 2 bootstrap trials used the Harris [6] sub-asymptotic EV model for the source distribution to evaluate convergence towards the Type 1 asymptote with increasing

$\Pi$ or $r$ for the various degrees of curvature expressed by the shape, $w$. To satisfy the many advocates of the GEV distribution that this study has been fair and comprehensive, Phase 3 investigates the reliability of the epoch methods operating on samples drawn from the GEV asymptote for various values of shape, $\xi$.

A complication arises in making the GEV parameters dimensionless in the manner of (4) and (5) to allow direct comparison with the Phase 1 results. The GEV location parameter $\langle \mu \rangle \to \Pi$, but the scale and shape parameters do not trend towards constants. The complication is resolved by bootstrapping the ensemble average GEV parameters: location $\langle \mu \rangle$, scale $\langle \sigma \rangle$ and shape $\langle \xi \rangle$; that were obtained for the various combinations of $\Pi$ and $w$ in Phase 1 and scaling the predictions by the corresponding dispersion to $V/D$, as before. This allows the Phase 3 results to be compared with the Phase 1 results using the same graphical formats, where $\Pi$ and $w$ serve to index the GEV parameters.

### 3.3.2. Source and Fitted GEV Parameters

Figure 17 shows how reliably the source GEV parameters are recovered by $10^4$ bootstrap trials of the GEV model. The range of GEV source parameters bootstrapped is indicated by the abscissa scales. The ensemble means of the location in (a) and scale in (b) are almost indistinguishable from the source values, but the shape in (c) shows a clear bias towards negative values, i.e., to Type 3 behavior, which decreases with increasing $R$. The standard error of location in (d) decreases with increasing $R$ and increasing source $\mu$. The standard error of scale in (e) collapses reasonably to a stable value for each $R$, as does the standard deviation of shape, $\sigma(\xi)$, in (f). (The standard error $\sigma(\xi)/\langle \xi \rangle$ is not presented because the mean passes through zero and introduces a singularity.)

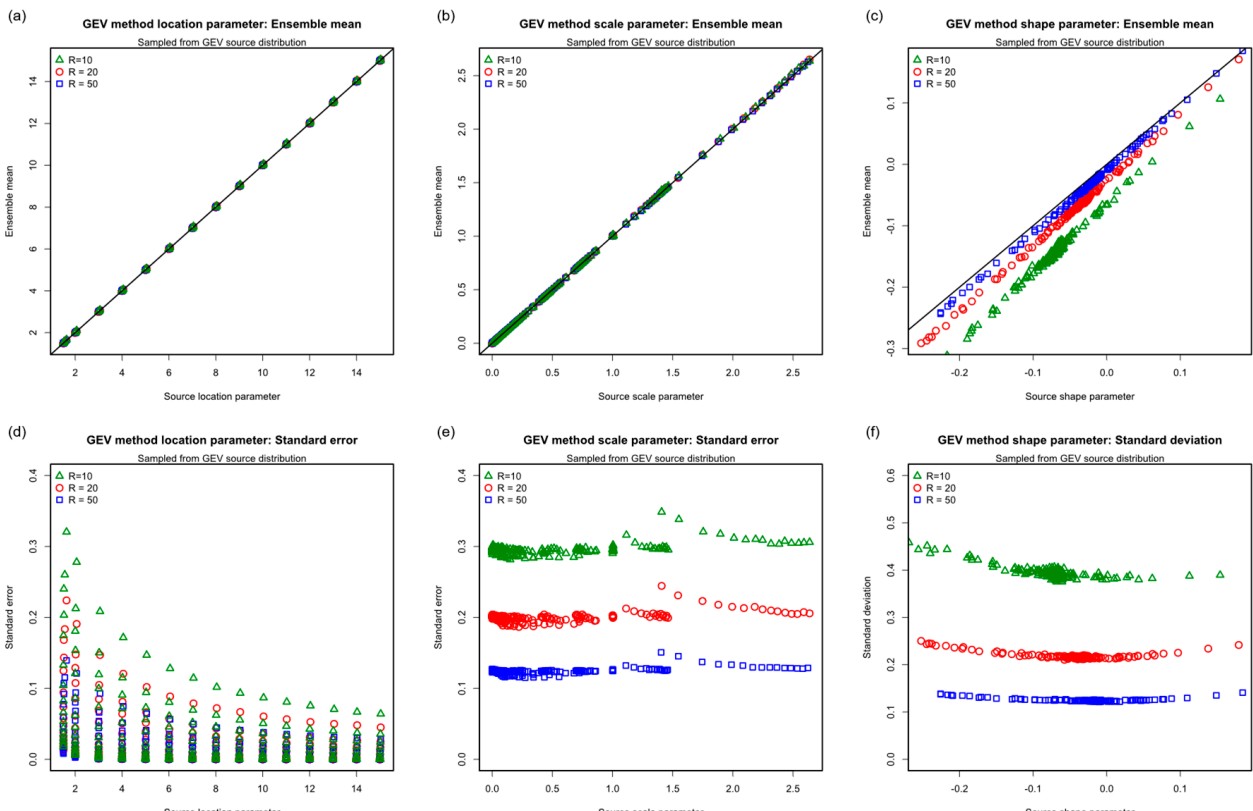

**Figure 17.** Quantile–quantile plots of GEV method parameters sampled from GEV source: (**a**) Location mean, $\langle \mu \rangle$; (**b**) Scale mean, $\langle \sigma \rangle$; (**c**) Shape mean, $\langle \xi \rangle$; (**d**) Location standard error, $\sigma(\mu)/\langle \mu \rangle$; (**e**) Scale standard error, $\sigma(\sigma)/\langle \sigma \rangle$; (**f**) Shape standard deviation, $\sigma(\xi)$.

### 3.3.3. Bias Errors

Figure 18 shows the mean bias errors of each method at MRI = 50 and 10,000 for $R = 50$ in the same format as Figures 4 and 5, where the abscissa $\Pi$ is directly equivalent to $\mu$ and the ordinate $w$ is an index to $\xi$. Here, (a) to (c) are directly comparable with Figure 4a–c, and (d) to (f) with Figure 5a–c. Despite the fundamental difference in source distributions, the Phase 3 bias errors are closely similar to Phase 1 in both form and value. The principal difference in (a) and (d) for the Gringorten method using source $w$ is that the contours now have an organized pattern reflecting the mismatch between the source GEV and the Gringorten sub-asymptotic model. The distinctive pattern for Gringorten fitting $w$ of Phase 1 is reproduced in (b) and (e) with only small differences in value. It would be expected that the results of fitting the GEV to samples from a GEV source would naturally be better, but improvement is confined to $\Pi < 3$, otherwise the bias errors are closely comparable.

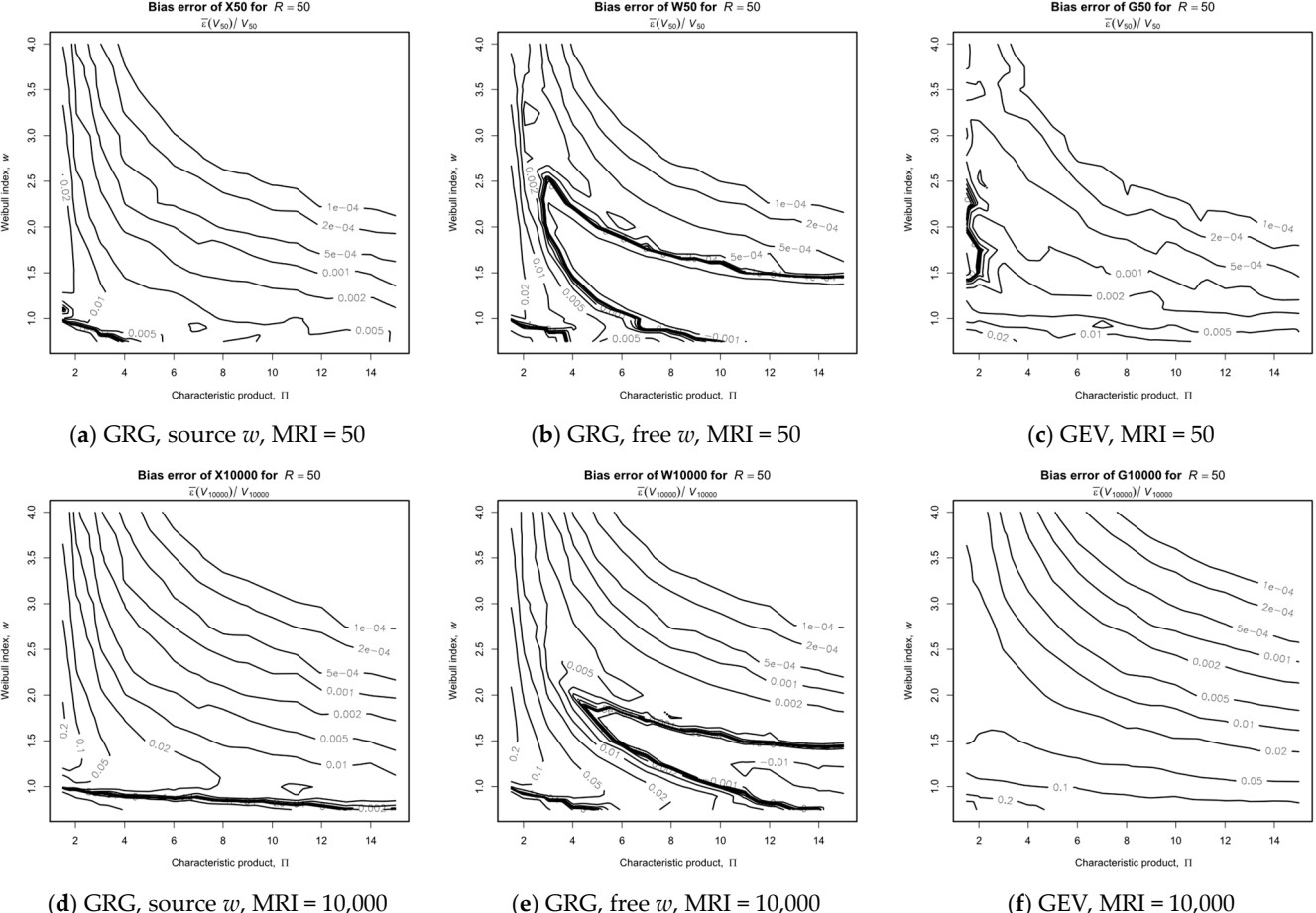

**(a)** GRG, source $w$, MRI = 50

**(b)** GRG, free $w$, MRI = 50

**(c)** GEV, MRI = 50

**(d)** GRG, source $w$, MRI = 10,000

**(e)** GRG, free $w$, MRI = 10,000

**(f)** GEV, MRI = 10,000

**Figure 18.** Mean bias error of epoch method predictions sampled from GEV source, $R = 50$.

### 3.3.4. Standard Errors

Figure 19 presents the corresponding standard errors. Here, (a) to (c) are directly comparable with Figure 6a–c, and (d) to (f) with Figure 7a–c. A comparison with the Phase 1 results shows them to be virtually identical.

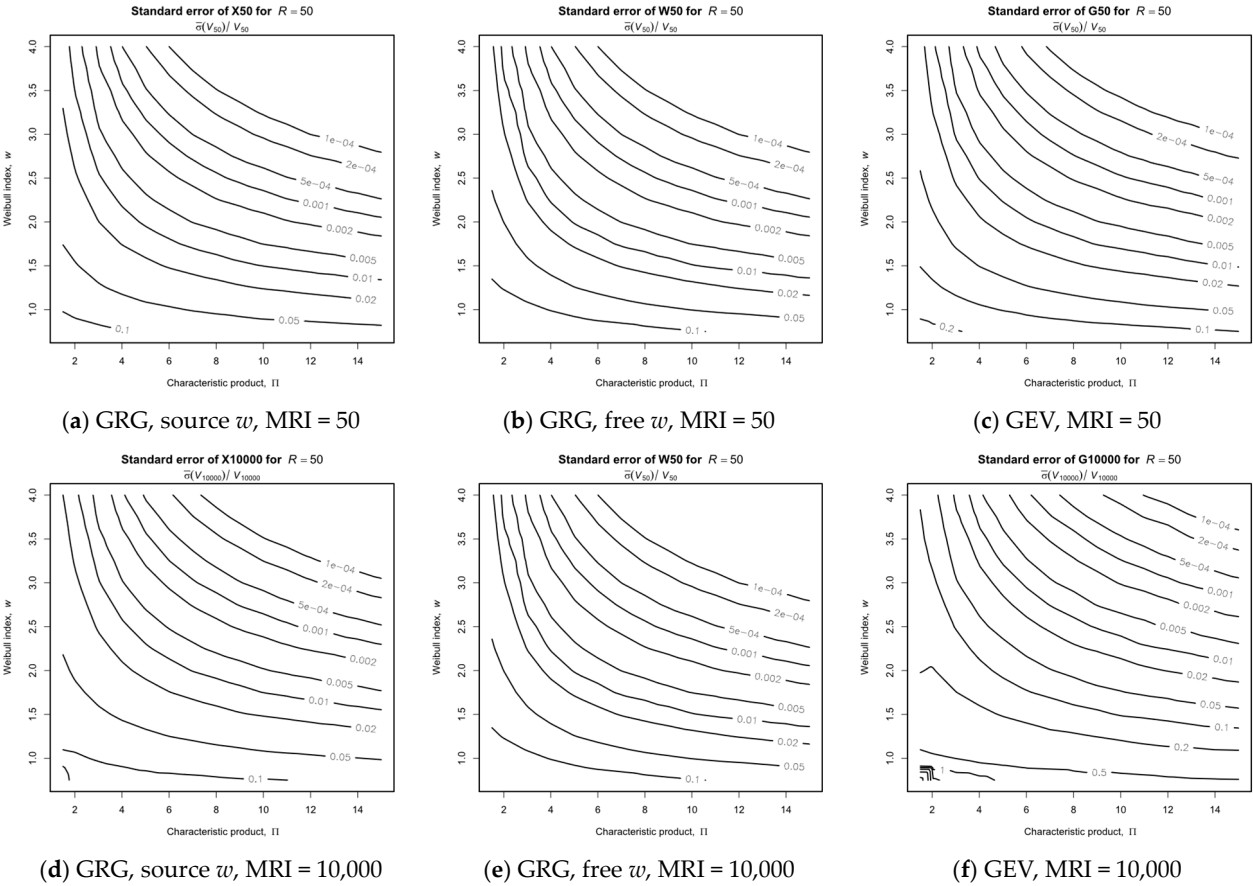

(**a**) GRG, source *w*, MRI = 50     (**b**) GRG, free *w*, MRI = 50     (**c**) GEV, MRI = 50

(**d**) GRG, source *w*, MRI = 10,000     (**e**) GRG, free *w*, MRI = 10,000     (**f**) GEV, MRI = 10,000

**Figure 19.** Standard error of epoch method predictions sampled from GEV source, $R = 50$.

### 3.3.5. Performance Overview

Figure 20 presents the QQ plots for $R = 20$ epochs at mean recurrence intervals MRI = 50 and 10,000 which are directly comparable with Figure 8a,b. Again, the results are closely similar to Phase 1, the principal difference being that the GEV method standard error is reduced by about 25% at MRI = 10,000, i.e., in the far tail where the GEV and sub-asymptotic Type 1 models differ the most.

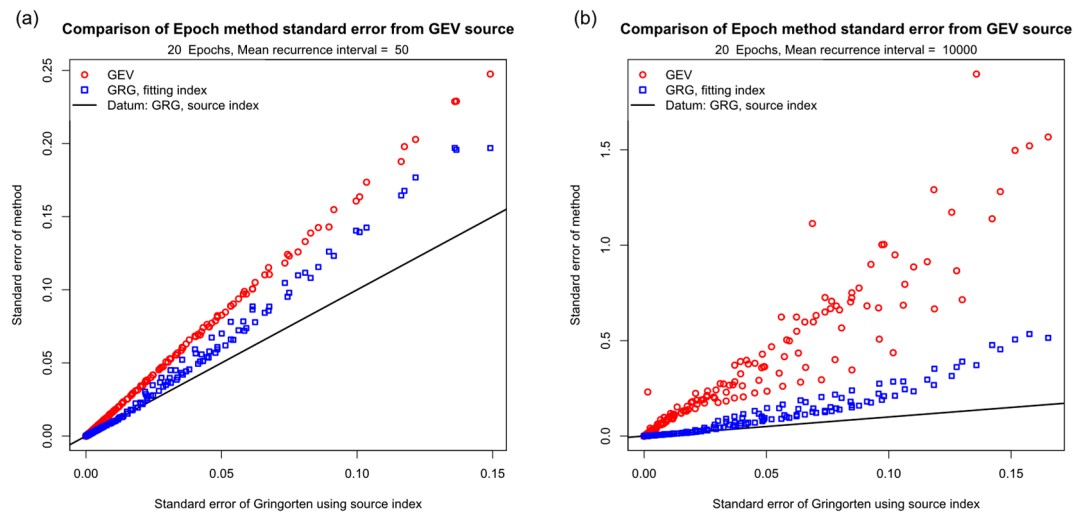

**Figure 20.** Quantile-quantile plots of standard errors of epoch method predictions sampled from GEV source, against Gringorten using the known shape datum: (**a**) MRI = 50; (**b**) MRI = 10,000.

The counter-intuitive conclusion from the Phase 3 trials is that predictions from the asymptotic GEV distribution are more reliably represented empirically by the sub-asymptotic Type 1 model than by the GEV over the range of parameters typical of wind speed observations. This shows that the GEV/GPD is inherently less reliable. The $V^w$ transformation defines a unique second derivative at any given wind speed. With the GEV, the same second derivative value can be obtained from multiple different combinations of GEV parameters, leading to higher variance. The GEV with $\xi < 0$ will only work well when the record length is long enough to approach $V_{max}$ [4].

## 4. Discussion and Prospects

This study is based on the presumption that mixed wind speed observations have been separated into individual disjoint components, or that the POT threshold is sufficiently high to pass only the most dominant component. There are procedures for fitting mixture distributions, i.e., to fit the Gomes and Vickery [16] model, $\Phi = \prod_i(\Phi_i)$, directly to the observations, but the proliferation of model parameters leads to larger variance and less reliable predictions.

The principal arguments of advocates for the GEV as a model for wind speeds are that there "must" be an upper limit to wind speed and that empirical fits of observed wind speed to the GEV consistently give negative values of shape, $\xi$. One argument reinforces the other in a circular fashion. This study shows the mean bias to negative $\xi$ occurs naturally through the action of statistical variance, even when the source distribution is GEV. However, the apparent limit, $V_{max} = \mu - \sigma/\xi$, for observed wind speeds is seen to rise with increasing record length, $R$, as more observations are obtained, whereas a real limit would remain constant. The negative bias of the GEV shape factor in Figure 17c, which decreases with increasing $R$, contributes to the observed trend for the estimated limiting wind speed to increase as more wind speed observations are collected. However, this is insufficient to account for the negative values of $\xi$ quoted in the literature, e.g., $\xi = -0.16$ quoted by Holmes and Moriarty [18] for a GPD analysis of downbursts at Moree, Australia. This is principally due to the difference between the sub-asymptotic Type 1 and asymptotic GEV/GPD models, where Figure 10c,d show will occur when $w > 1$ and $\Pi$ is small.

The sub-asymptotic Type 1 model is exact when the parent is a Weibull distributed iid process or is tail-equivalent to Weibull over the range of top ranks, $M$, [8,19]. In which case, the wind speed raised to the power of the Weibull index, $V^w$, becomes exponentially distributed, and extremes converge extremely rapidly to the Gumbel distribution with increasing $r$ [2]. When Weibull tail-equivalence is doubted, the best fit for index, $w$, can simply be regarded as a pragmatic way of obtaining the fastest convergence. The Weibull distribution has long been used empirically for the parent distribution of wind speeds and is a good empirical model for wind speeds in climates dominated by temperate depressions, where the index lies in the range $w$ = 1.8 to 2.2. Its ubiquity has meant it is often misused as a single distribution to represent mixtures, resulting in unrealistically low index values as in Figure 11a, or else replaced by other single distributions that give a better empirical fit. A review of 35 previous studies [13] concluded that the quest for a universal single distribution for wind speed is futile—instead, the individual components of the mixture should be separated from the mixture and each separately assessed.

The practical application of Weib-XIMIS, shown here as the most reliable method for assessing extreme wind speeds, hinges on the separation of the mixture into individual components and the determination of $w$ for each component. If separation is not possible, then a high POT threshold will pass only the dominant component, but this still requires the corresponding relative frequency to implement the procedure in Figure 11a. As a fallback position, the second most reliable method is the three-parameter XIMIS fit which is less satisfactory but still more reliable than GPD.

The OENM model [7,13] was developed for mixtures of components that are not constrained by direction and are therefore random in orthogonal horizontal axes. It is also seen to perform well for weakly directional components [13,20,21], e.g., for monsoons,

and for some moderately directional components, such as sea and land breezes (as these generally have a random component parallel to the coast, albeit weaker than the principal component normal to the coast), at both local and continental scales. However, OENM is poor for components that are tightly constrained in direction by orography, e.g., the strong katabatic winds of Antarctica [13]. Each OENM component is represented as a bivariate normal distribution, which resolves on orthogonal horizontal axes as elliptical contours offset from the origin. When the ellipticity and offset are both zero, Davenport [22] showed that the transformation into polar coordinates creates a distribution of wind speed that is exactly Weibull with $w = 2$ (Rayleigh). Harris and Cook [7] noted that variations in ellipticity and offset account for the observed departures from this ideal and conclude that the Weibull distribution is an effective surrogate for the OENM marginal distribution of wind speed in the range $w$ = 1.8 to 2.2. Reference to Figure 16 shows that this $\mp 10\%$ variation around $w = 2$ limits the prediction errors when $\Pi = 2$ to $\pm 2\%$ at MRI = 50 and $\pm 5\%$ at MRI = 10,000, decreasing by a factor of 10 each time $\Pi$ is doubled. This suggests that a fixed value of $w = 2$ may be adequate for directionally unconstrained components.

Recently, methods have been developed to isolate and automatically separate mesoscale components [23–27], such as gust fronts and thunderstorms, from the macroscale geostrophically driven wind components, which are a great improvement on earlier manual methods. (See Additional Bibliography and the literature review in [27].) Unfortunately, some recent methods rely on observational data for which there is no long-term archive, e.g., satellite data. Others require data observed at shorter intervals than the standard WMO FM-12 (SYNOP) and FM-15 (METAR) reports. An exceptional dataset is the ASOS archive of observations at 1-min intervals from 2000 onward from almost 1000 active meteorological stations across the contiguous USA (CONUS). Chen and Lombardo [24], Solari et al. [25], and Cook [27] have tested separation methods on ASOS data, with the latter succeeding in separating out a larger range of wind components than the other methods.

This study complements and reinforces the previously published but contested theoretical and statistical arguments that GEV and GPD should be replaced by the sub-asymptotic Type 1 model for the assessment of extreme wind speeds. It completes the last of a series of tasks required in preparation for a comprehensive EV analysis of the extreme wind speeds across the CONUS. The earlier tasks were:

1. Locating the ASOS anemometers with good (WMO Class 1 or 2) exposures [28];
2. Curating the ASOS data to detect, remove or repair errors and artefacts [29] for these sites;
3. Classifying all gust events exceeding 20 kn [27] and separating into disjoint components by causal mechanism; and
4. Determining the effect that the "Test 10" ASOS real-time quality control algorithm has in erroneously culling valid observations since its introduction in 2014, and the impact this has on the assessment of extreme gusts [30].

Prospects for the future include EV analysis of the separated ASOS convective gust components from [27] using the Weib-XIMIS method proposed in this study with the aim of mapping gust speeds across CONUS for each individual component, and OENM analysis of the synoptic-scale components. Such studies should indicate whether the Weibull shape, $w$, is consistent for each component class, e.g., isolated thunderstorm downbursts (Class 5 in [27]), and whether this matches the OENM expectation of $w \approx 2$. A parallel analysis of the ASOS data by GEV/GPD is also anticipated as the sub-asymptotic Type 1 model is unlikely to be widely accepted until such work is completed and peer-reviewed.

### 5. Conclusions

- Peak-over-threshold methods are shown to always be more reliable than epoch methods due to the additional sub-epoch data.
- Predictions from the generalized asymptotic methods are always less reliable than those from the sub-asymptotic methods by a factor that increases with the mean recurrence interval.

- These conclusions reinforce the previously published theoretical and statistical arguments for using the sub-asymptotic Type 1 model and against using the GEV/GPD for assessing extreme wind speeds.
- A new two-step Weibull-XIMIS hybrid sub-asymptotic method is shown to have superior reliability.

**Supplementary Materials:** The following supporting information can be downloaded at: https://www.mdpi.com/article/10.3390/meteorology2030021/s1 or https://doi.org/10.17632/cxfgnfjvn3.1. They comprise: (1) An Additional Bibliography of papers that have influenced this study but are not directly cited in the References. (2) R data files of the ensemble mean and variance of all model parameters and predictions. (3) R scripts to replicate or extend the R data files. (4) R scripts to chart the results for various combinations of parameters. (5) A PDF giving the keys to the R data columns and instructions for running the scripts.

**Funding:** This research received no external funding.

**Data Availability Statement:** All data used in this study are available in the Supplementary Materials.

**Conflicts of Interest:** The author declares no conflict of interest.

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
