# Peer review of "Reliability of Extreme Wind Speeds Predicted by Extreme-Value Analysis"

_2674-0494, doi:10.3390/meteorology2030021_

Round 1
Reviewer 1 Report
The work shows an analysis of the models available to describe the extreme wind data distribution. It compares asymptotic generalized extreme value (GEV) and pareto distributions (GPD) with sub-asymptotic distribution models. The author concludes that the sub-asymptotic models should be used for extreme wind assessment. The author also discusses the importance of knowing the distribution shape parameter and proposes a hybrid sub-asymptotic model where the shape value can be pre-determined.
I think that the manuscript is mainly targeted for an audience with experience in the extreme wind data distributions and extreme-value analysis. I would strongly encourage the author to provide additional references describing the background information. I found hard to find that information with the list of provided references.
I think that two issues, as explained below, should be addressed before publication:
1) I think that the author should refrain from saying that the “new two-step Weibull-XIMIS hybrid method is shown to have superior reliability”. In my opinion there is not enough evidence supporting this statement since the model has not been compared with respect to the others. Figs. 14 and 15 only show a comparison amongst different “flavors” of the XIMIS model.
2) The comparisons amongst the different models (for example Fig 8 and 9) are shown for R = 20 and, if I understand correctly, this number represents the number of years of the data used. Why does the author choose R = 20? Would the conclusions hold for different (maybe lower) values?
Few minor issues:
- Fig. 11 has ln on both axes, whereas Fig 12 has log. I think this should be adjusted.
- The title of Fig 17(f) shows “Standard deviation” but I think it should be standard error for consistency with the other plots.
Author Response
I think that the manuscript is mainly targeted for an audience with experience in the extreme wind data distributions and extreme-value analysis. I would strongly encourage the author to provide additional references describing the background information. I found hard to find that information with the list of provided references.
> A certain degree of experience is required to understand any scientific paper and a balance must be struck by the author. The paper is indeed targeted to a readership with an interest in extreme winds and therefore having some degree of experience. The citations point to the key reference to the statement being made and they lead naturally to discussion and further references to be followed. The Additional Bibliography, which the reviewer may have missed, provides most of the references required. With this, I believe that the correct balance has been struck.
I think that two issues, as explained below, should be addressed before publication:
1) I think that the author should refrain from saying that the “new two-step Weibull-XIMIS hybrid method is shown to have superior reliability”. In my opinion there is not enough evidence supporting this statement since the model has not been compared with respect to the others. Figs. 14 and 15 only show a comparison amongst different “flavors” of the XIMIS model.
> The study compares the reliability of the sub-asymptotic Gringorten epoch and XIMIS POT methods with the asymptotic GEV and GPD methods. The conclusion is based on the whole study, not just Figs 14 and 15 which only compare preconditioning options for XIMIS.
2) The comparisons amongst the different models (for example Fig 8 and 9) are shown for R = 20 and, if I understand correctly, this number represents the number of years of the data used. Why does the author choose R = 20? Would the conclusions hold for different (maybe lower) values?
> First mention of R = 20 in L166 relates to the variance of “a small number of samples” masking any systematic curvature and making its estimation difficult. By implication, masking would be worse for smaller R values. Second mention of R = 20 in L187 is in setting the calibration values of R = 20 and 50 to match the typical range of observed hourly wind speed observations. Note “hourly” inserted in L188.
Few minor issues:
- Fig. 11 has ln on both axes, whereas Fig 12 has log. I think this should be adjusted.
> Yes, done.
- The title of Fig 17(f) shows “Standard deviation” but I think it should be standard error for consistency with the other plots.
> No, you missed the footnote – now L488-489 – “(The standard error is not presented because the mean passes through zero and introduces a singularity.)“. Figure 17 caption corrected to “(f) Shape standard deviation, σ(ξ)".
Reviewer 2 Report
Reliability of Extreme Wind Speeds Predicted by Extreme-value Analysis
By Nicholas John Cook
General comments.
This paper addresses different methods for EV analysis. The author concludes that the reliability of the classic asymptotic methods is less in comparison with sub-asymptotic methods, mainly for increasing MRI. The author has a substantial experience on the paper’s subject and this paper is one in a series. I guess this paper is worth publishing after some text revisions I point below.
Text revisions
Line 91,
The Author should stand the meaning of POT (peak-over-threshold) in the Introduction, for the sake of clarification.
Line 101,
Then the R POT values are unevenly distributed among the R epochs so that, 100 on average 37% of the epochs will contain no value and the other 67% may contain more 101 than one value, i.e., the 2nd or 3rd highest values in those epochs.
Did not the author mean “33 %”?
Line 110,
The special case where ? = 0 is the Type I or Gumbel distribution, which is unlimited in both tails.
Please change Type I by Type 1.
Line 255,
The show the significant advantage of knowing the shape in 254 advance and so fitting only for location, ?, and scale, D.
Did not the author want to mean “They show”?
At some point, the author becomes to use names of authors as reference, instead of their numbers in the References. I guess the standard of MDPI is to use numbers.
Line 534,
This study shows the mean bias to negative ? occurs 533 naturally the through the action of statistical variance, even when the source distribution 534 is GEV.
Please, get rid of this underlined “the”.
Author Response
Text revisions
Line 91 – The Author should stand the meaning of POT (peak-over-threshold) in the Introduction, for the sake of clarification.
> Yes, done.
Line 101 – Then the R POT values are unevenly distributed among the R epochs so that, 100 on average 37% of the epochs will contain no value and the other 67% may contain more 101 than one value, i.e., the 2nd or 3rd highest values in those epochs. Did not the author mean “33 %”?
> No, the 67% was wrong, corrected to 63%. 1/e = 0.36788 = 37%. 1-1/e = 63%.
Line 110 – The special case where ? = 0 is the Type I or Gumbel distribution, which is unlimited in both tails. Please change Type I by Type 1.
> Yes, done.
Line 255 – The show the significant advantage of knowing the shape in 254 advance and so fitting only for location, ?, and scale, D. Did not the author want to mean “They show”?
> Yes, done.
At some point, the author becomes to use names of authors as reference, instead of their numbers in the References. I guess the standard of MDPI is to use numbers.
> L83, L353, L354, 462 instances found and corrected,
Line 534 – This study shows the mean bias to negative ? occurs naturally the through the action of statistical variance, even when the source distribution is GEV. Please, get rid of this underlined “the”.
> Yes, done.
Reviewer 3 Report
The paper is good. It is very well written and the presentation is very analytic and goes in depth. Although the present reviewer is in favour of the GEV - epoch methodologies however, the approach presented in this paper is of equal interest and merit.
I have not much to add in my remarks. Some suggestions that will probably facilitate the reading of the paper and the understanding of the ideas presented are the following.
1. The presentation of the numerical results is very lengthy. I would strongly suggest to summarize (maybe in the Introduction or in an extra section) the different bootstrap trials: what is the subject of each trial, which are the underlying conditions and the expected results.
2. Since the numerical results cover more than 15 pages, I would suggest to include in the discussion section a summary of the most important numerical results. This could be very helpful for the non-familiar reader.
3. In some figures (e.g., 8, 14, etc.) it could be helpful to provide the slopes of the best-fit lines in order to have a quantification of the differences between the examined approaches even through this very simplistic approach.
4. For the results shown in Figure 7 (for MRI=10000), provide also some very short comments.
5. There are very different approaches for the selection of the POT values in real applications. Most of them are based on the assumption that the selected values are (or should be) eventually independent. For instance in the Introduction it is stated that r=150 to r=300, roughly corresponding to 29 - 58 hours between events. However, this is not always the case. Please provide some more detailed comments as regards the selection of the POT values.
Very minor typos and errors.
Author Response
The paper is good. It is very well written and the presentation is very analytic and goes in depth. Although the present reviewer is in favour of the GEV - epoch methodologies however, the approach presented in this paper is of equal interest and merit.
> GEV in this context is an empirical fit using the wrong asymptote which mimics incomplete convergence to the correct asymptote. This study shows that the GEV is also less reliable than the sub-asymptotic method, particularly on extrapolation to large MRI.
I have not much to add in my remarks. Some suggestions that will probably facilitate the reading of the paper and the understanding of the ideas presented are the following.
- The presentation of the numerical results is very lengthy. I would strongly suggest to summarize (maybe in the Introduction or in an extra section) the different bootstrap trials: what is the subject of each trial, which are the underlying conditions and the expected results.
> My view is that the presentation of results, although lengthy, is already a summary of the extensive analysis results provided in the Supplementary Materials. Further summarization is in danger of trivialising the issues.
- Since the numerical results cover more than 15 pages, I would suggest to include in the discussion section a summary of the most important numerical results. This could be very helpful for the non-familiar reader.
> A certain degree of experience is required to understand any scientific paper and a balance must be struck by the author. The paper is indeed targeted to a readership with an interest in extreme winds and therefore with some degree of experience. The discussion section summarises and discusses the most important conclusions drawn from the numerical results.
- In some figures (e.g., 8, 14, etc.) it could be helpful to provide the slopes of the best-fit lines in order to have a quantification of the differences between the examined approaches even through this very simplistic approach.
> The aim of these figures is to show how the free fit to the model shape increases the standard error in comparison with the known shape datum. The best fit lines have no relevance to this aim, especially as the scatter is not wholly random, but is a collection of individual curves that depend on the source shape parameter. The information requested by the reviewer is effectively that presented graphically in Figure 9.
- For the results shown in Figure 7 (for MRI=10000), provide also some very short comments.
> See new L245-249.
- There are very different approaches for the selection of the POT values in real applications. Most of them are based on the assumption that the selected values are (or should be) eventually independent. For instance in the Introduction it is stated that r=150 to r=300, roughly corresponding to 29 - 58 hours between events. However, this is not always the case. Please provide some more detailed comments as regards the selection of the POT values.
> The reviewer conflates two separate, but linked, issues.
The r=150 to 300 range for the annual rate of independent events in synoptic windstorms and the lower values for mesoscale events is quoted to indicate that convergence to the ultimate asymptote will not be achieved (Galambos and Macri [4] suggest r=100000 is required for convergence).
The selection of POT values merely requires independence between successive values, so their rate can be any value smaller that the rate of all independent values. Selection may be by threshold value, minimum separation time, or by the identification of individual meteorological events, e.g., thunderstorms. All practical methods imply some small degree of convergence from the parent towards the extreme. There is a clear hierarchical progression: correlated parent → correlated local peaks → independent local peaks → epoch maxima. This is addressed in numerous papers including some in the Additional Bibliography. It is not in the limited scope of this study to address this further.
Author Response
> The reviewer has debated the issues raised in the study with a degree of well-informed scepticism. for which I am grateful. Apart from my sometimes collaborator Mr Harris (who takes a more fundamental theory-based view on the debate, whereas I take a more pragmatic data-based view) I have few opportunities to debate with others with similar expertise and experience, so I am grateful for this. I have responded to each of the comments which are, effectively, a continuation of the debate and would be more appropriate as a discussion contribution.
Readership – I doubt that this manuscript is suitable for the intended readership of Meteorology.
> This is not the view of the commissioning editor or the other three reviewers.
It addresses methodology of extreme value analysis intended for wind speed data, and is based entirely on simulated data, with no analysis of data collected in the field.
> Obviously, because with real field data there is no “correct” known results to compare against. This is a calibration exercise, where it is as helpful to find there are “modest differences” as to find there are large differences in performance. The study fills a gap in our knowledge of how reliably each method would assess a single record of observed data.
Aims and results – Goal is to compare the performances of GPD/GEV-based return value estimators and of Weibull-based return value estimators, and secondarily, of estimators based on POT and on epoch maxima. The author concludes that the GPD/GEV-based return value estimators are always less reliable than the Weibull-based return value estimators, by a factor that increases with mean recurrence interval. However, this conclusion cannot be drawn from the simulations performed, as these simulations (except in Phase 3) all sample from an exact Weibull distribution, so methods designed specifically for the Weibull distribution will in general perform better than more generic methods.
> The reviewer misses the principal point, that the performance of the four methods are compared using the same source data in every trial. Each method would be expected to faithfully represent extremes drawn from a Weibull parent when r > 10 (to satisfy the Cauchy approximation on which the asymptotic models are founded). The sub-asymptotic model does this by accelerating convergence. GEV/GPD does this by assuming full convergence to an incorrect asymptote which mimics non-convergence. The “modest differences in performance” noted below by the reviewer apply only to the mean bias and the standard errors at the “design” MRI=50. At the “critical infrastructure” MRI=10000, the difference in standard errors between the methods are substantial.
Even in phase III, the GPD’s sampled from are estimates from the same Weibull data, so these samples are still close to the underlying Weibull distributions (they are locally similar, which results in similar estimates).
> The reviewer is wrong on this point. Phase III generates extremes directly from the GEV distribution using the GEV parameters with the empirical ensemble values that correspond to the Weibull parameters in earlier phases. This is just a device to enable normalisation of the results and allows direct graphical comparison with the earlier Phases. No Weibull data are used in Phase III.
Considering samples from the Weibull distribution is sufficient to demonstrate that GPD/GEV based methods are biased on wind speed distributions, because the latter are overall similar to Weibull distributions, and for these, convergence to the GPD or GEV tail limits is slow. This issue has been addressed earlier in a number of excellent publications by the author and dr. Harris. However, to prove empirically that the Weibull-based methods are better than GPD/GEV based methods for wind speed data, superior performance should be demonstrated on data from a sufficiently broad class of distribution functions that it can be reasonably assumed to represent the variation among wind speed distributions in nature (in particular, exhibiting realistic deviations from exact Weibull tails).
> Here, “better” means more reliable in the sense of smaller standard error. I disagree with the referee’s argument in this respect. Provided the issue of mixed distributions has been properly addressed, “realistic deviations from exact Weibull tails” are for more likely to be due to the greater statistical variance of the top ranks that by the real physical mechanism not having Weibull equivalence. I have yet to find a single (not mixture) wind speed distribution that does not qualify. Yes, the sub-asymptotic model is exact for a Weibull parent, so is fairly described as “Weibull based”, but it is applicable to all distributions in the FT1 domain of attraction for which convergence is accelerated by the transformation Vw (see later comment).
The latter is difficult, because what should be included in this class? One might consider various types of Weibull mixtures. However, even with these, the problem remains that the study outcomes will be sensitive 1 to the choice of the mixtures to consider (by design, one can make it easy, or very hard, for a Weibull-based estimator or for a GPD-based estimator).
> Unreasonable to consider any mixture of disjoint components as a single distribution. The components must first be separated, then we are back to considering individual Weibull parents. I have yet to find a set of wind observations that does not survive the null hypothesis test for Weibull tail equivalence. However, there is the prospect with the ASOS data of treating thunderstorm downbursts as a class, represented by a 450 location “superstation” providing around 9000 station-years of data, and sampling shorter observation periods from this.
In any case, simulations based only on data sampled from Weibull distributions are not conclusive.
> Line 78. “Even if this were not the case, EV models should faithfully represent extremes sampled from this model.”
In particular the proposed methods for estimation of the index from the full sample or part of the sample and the estimation of ”effective” sample size N from a POT sample would be sensitive to departures from the exact Weibull shape. In this respect, a major weakness of the examined Weibull-based methods is that they do not specify in what sense the Weibull tail is supposed to approximate the true tail of a wind speed distribution (it is only stated that wind speed tails are similar to Weibull tails).
> The role if the index in (2) is to accelerate convergence. Applied to real observations, the observed POT distribution is not constrained to Weibull equivalence, so the EV distribution on a Gumbel plot may be a curve instead of the anticipated straight line. In practice, a straight line (Type 1) fit to Vw always survives a null-hypothesis test provided, of course, that a mixture distribution has been separated into individual components. Also note in the discussion (L555-556) “When Weibull tail-equivalence is doubted, the best fit for index, w, can simply be regarded as a pragmatic way of obtaining the fastest convergence.”
On the other hand, the GPD and GEV, for all their limitations, are tail approximations in a specific sense, valid under a fairly weak regularity assumption. How accurate such an approximation is for a particular distribution or data sample then needs to be assessed, of course (e.g. assessment of bias by examining threshold dependence plots, assessment of variance, etc.). But one cannot easily claim that the estimator is invalid for a particular variable. In fact, alternative regularity assumptions have been proposed for the purpose of improving extrapolation to high return periods, which would be more suitable for Weibulllike wind speed distributions than the classical tail limit: the Weibull tail limit (e.g. [4]) and the Generalized Weibull (GW) tail limit ( [2]) (with shape equivalent to a Weibull tail with offset). Tail quantile estimators based on these assumptions (or for the ”log-GW” tail limit, which is the GW tail limit of the logarithm of the random variable) are presented in the references in [4], [3] and [1]. Furthermore, given the relatively modest differences in performance between Weibull and the generic GPD/GEV based methods shown, it cannot be claimed that the generic methods have been invalidated by this study.
> This author does not seek to “invalidate” GEV/GPD in this study – this has been previously argued and disputed elsewhere. This study only seeks to show that sub-asymptotic Type 1 methods are more reliable than the generalized asymptotic methods when applied to extremes from tail-equivalent Weibull parents. However, the author believes that, taken with the asymptotic considerations expressed elsewhere, the results are sufficiently persuasive to prefer the sub-asymptotic approach. The generalized methods assume full convergence to the asymptote and r is too small for this to be valid.
They have somewhat higher variance, which one might be willing to accept as the price to be paid for weak assumptions. Of course, bias would be more of an issue for POT-based estimates based on lower wind speed thresholds, but these are not considered in the simulations.
> What? At MRI=10000 (critical infrastructure) you would be content with 50% standard error from GEV/GPD as opposed to 10% from XIMIS operating on the same data? The corresponding factors on wind loads are 2.25 for GEV/GPD as opposed to 1.21 for Weib-XIMIS. Coupled with a bias towards underestimation. The price is too high as it exceeds design safety factors.
Additional comments
- I would encourage the author to submit his analysis of the ASOS archive of observations. In my view, this should not require a ”stamp of approval” of his Weibull-based methods by demonstration of their superiority over alternatives.
> I agree, but not all commentators would. A “stamp of approval” is always helpful.
A simple way to circumvent criticism would be run a parallel analysis using the GPD and compare the estimates from the different methods, which would also be of interest in itself (e.g. to show systematic differences with interpretation). Another option would be to try an estimator based on the Weibull-tail limit or GW tail limit mentioned above.
> This study is a preliminary step towards this reviewer’s suggestion.
- The term bootstrapping is commonly reserved for determining the error statistics of estimates by methods based on random sampling. It is only applied in Phase 3 (on the GPD estimates from Weibull data). The rest of the study is based on simulation.
> I would agree with this for non-parametric bootstrapping, i.e., the estimation of the error statistics from the observed data. However, this study uses parametric bootstrapping, i.e., sampling from a parametric model, where the distinction between “bootstrapping” and “simulation” is blurred. Phase 3 is not fundamentally any different than Phases 1 and 2 except for the choice of parametric model. All three phases seek to determine bias and standard error by Monte Carlo sampling, so the description as “bootstrapping” is appropriate. I reserve the term “simulation” for the generation of synthetic time-series data.
Round 2
Reviewer 4 Report
I concede that suitability for the journal is to be decided by the editor, so I drop this point.
From the author's response, I have the impression that the main scientific issues raised by me have not been completely understood. Therefore, I maintain my previous conclusions and advice.